# Why is constrained neural language generation particularly challenging?

**Cristina Gârbacea**  *garbacea@uchicago.edu*
*Data Science Institute*
*University of Chicago*

**Qiaozhu Mei**  *qmei@umich.edu*
*School of Information, Department of EECS*
*University of Michigan*

**Reviewed on OpenReview:** *https://openreview.net/forum?id=Vwgjk5ysWn*

## Abstract

Recent advances in deep neural language models combined with the capacity of large scale datasets have accelerated the development of natural language generation systems that produce fluent and coherent texts (to various degrees of success) in a multitude of tasks and application contexts. However, controlling the output of these models for specific user and task needs is still an open challenge. This is crucial not only to customizing the content and style of the generated language, but also to their safe and reliable deployment in the real world. We present an extensive survey on the emerging topic of *constrained* neural language generation in which we formally define and categorize the problems of natural language generation by distinguishing between *conditions* and *constraints* (the latter being testable conditions on the output text instead of the input), present constrained text generation tasks, and review existing methods and evaluation metrics for constrained text generation. Our aim is to highlight recent progress and trends in this emerging field, informing on the most promising directions and limitations towards advancing the state-of-the-art of constrained neural language generation research.

## 1 Introduction

Recent advances in the field of natural language generation (NLG) (Gatt & Krahmer, 2018) have resulted in models able to produce realistic, coherent, and fluent texts in a multitude of natural language processing tasks. Powerful large scale language models can be readily used to perform unconditional language generation, however these models provide little control over attributes of the *generated* texts. Unlike conventional methods which were able to provide fine-grained control over many aspects of the system output including incorporating domain-specific dictionaries, terminology or certain words in the generated output, neural end-to-end approaches remove many of these knobs and switches (Post & Vilar, 2018). However, imposing constraints on the output generated by these models is crucial for achieving useful and safe language generation in a multitude of real world application scenarios. For example, it can help avoid generic and meaningless responses in dialogue systems (See et al., 2019), personalize dialogue agents based on user features that lead to more engaging and meaningful conversations (Zhang et al., 2018a), ensure non-offensive sentence completion and friendly communication (Liu et al., 2019b), intervene on the system output in interactive scenarios where domain specific terminology must be included in the generated texts (Crego et al., 2016), or aid in creative applications such as poetry generation or assisted story writing (Peng et al., 2018). Moreover, controlling a generic pretrained language model in order to satisfy certain desiderata helps avoid generating toxic content, prevents demographic biases, can steer generations towards a desired topic or style (Khalifa et al., 2021), and helps communicate intentions in suitable manners for different situations, target audiences and environments (Lample et al., 2018; Li et al., 2018). Incorporating prior knowledge and target

side constraints in text generative models has numerous applications in many natural language processing areas, including dialogue systems, machine translation, question answering, text summarization, text simplification, image captioning, etc. Unquestionably, constrained text generation is important in many real-world applications, but compared to other instances of natural language generation, constrained text generation using neural networks remains an open challenge.

We identify the following reasons why constrained neural text generation represents a much harder problem compared to other instances of neural text generation: *i) lack of model expressiveness*: current models are not expressive enough to incorporate arbitrary constraints, defined as testable conditions on the *output* text, into the objective function at training time; *ii) lack of suitable evaluation metrics*: while one can verify whether an output satisfies a constraint or not, it is usually hard to measure *to what extent* an output satisfies a constraint, and it is even harder to jointly evaluate this with other properties of the generated text (such as relevance or coherence); *iii) difficulty in constrained optimization*: even if constraints can be expressed and added to the objective function, they are usually non-differentiable, especially at the token level, which is bad since most methods model and generate text as a sequence of tokens; *iv) lack of constrained text generation datasets* that are diverse and representative enough of the variety of practical constraints.

Commonly used sequential text generation methods and architectures assume a rigid modeling of the output sequence based on word ordering, with tokens generated progressively one at a time in a standard left-to-right manner (Chan et al., 2019). Such autoregressive models cannot easily express constraints at arbitrary positions in the generated sequence or satisfy constraints involving multiple input objects. In addition to these issues, it is generally more challenging to incorporate multiple and heterogeneous constraints, which conform to given rules, topics, sentiments, lexical constraints, or pre-defined stylistic and content attributes.

Our work focuses on the emerging problem of neural natural language generation with constraints. We first define the problem and differentiate between the ambiguous use of *conditions* and *constraints* in natural language generation, including examples that represent instantiations of the constrained neural text generation problem. We then survey approaches, learning methodologies and model architectures employed for generating texts with desirable attributes, and corresponding evaluation metrics. We conclude with open research problems and limitations of current models. Our work aims to draw clear boundaries between the confusing terminology used in the neural language generation literature, highlight the main approaches and discuss how they suffer from the general challenges of constrained text generation, and serve as an informative guide for solving these general challenges and advancing meaningful, useful, and safe constrained NLG research.

## 2 Problem Definitions

We formally define the problem of natural language generation, accounting for context, conditions, and constraints placed on text generative models. First, we aim to articulate the key difference between condition and constraint since the distinction between these concepts is rather blurred in the natural language processing literature. Given a text generation task defined as $g(X) \to X'$, we define *condition* as a testable statement of the *input X*, and *constraint* as a testable statement of the *output X'*.

Accounting for the distinction above, we divide the text generation problem into three categories: *i) generic or free-text generation* which we present in Section 2.1, *ii) conditional text generation* which we introduce in Section 2.2, and *iii) constrained text generation* which we outline in Section 2.3. The focus of our work is on the particular problem of *constrained* text-to-text generation, leaving aside text generation tasks from other types of inputs such as data-to-text generation or image-to-text generation which are *conditional* in nature according to our definitions.

### 2.1 Generic/Free-Text Generation

The problem of generic text generation considers the intrinsic history of words generated until the current timestep in the sequence as context, and does not place any external user-defined conditions or constraints on the model output. We formally define it in what follows.

Given a discrete sequence of text tokens $\boldsymbol{x} = (x_1, x_2, \ldots, x_n)$ as input where each $x_i$ is drawn from a fixed set of symbols, generic text generation aims to learn the unconditional probability distribution $p(\boldsymbol{x})$ of sequence $\boldsymbol{x}$. This distribution can be auto-regressively factorized using the chain rule of probability (Bengio et al., 2003) into a product of conditional probabilities $p(\boldsymbol{x}) = \prod_{i=1}^{n} p(x_i|x_{<i})$ to perform density estimation and generation of text data. When $p(\boldsymbol{x})$ is modeled by a neural network with parameters $\theta$, the neural network is trained to minimize the negative log-likelihood $\mathcal{L}(D) = -\sum_{k=1}^{|D|} \sum_{i=1}^{N_k} \log p_\theta(x_i^k|\boldsymbol{x}_{<i}^k)$ over a collection of samples $D = \{\boldsymbol{x}^1, \ldots, \boldsymbol{x}^{|D|}\}$, where $N_k$ is the length of $\boldsymbol{x}_k$. To generate new samples, each token $x_i$ is iteratively sampled from $p_\theta(x_i|\boldsymbol{x}_{<i})$ and is fed back into the model as the input for the next timestep.

Large scale models for generic text generation show promising abilities to imitate the distribution of natural language and generate long-term realistic and coherent texts, however such free-text generation models place a lot of burden on the generative model to capture complex semantic and structural features underlying the data distribution; this can often result in repetitive, contradictory, and largely randomized generated texts (Holtzman et al., 2020). Notably, the content generated by free-text generative models cannot be controlled with respect to particular attributes and modes of the data distribution. This inability to control which regions of the data distribution are generated is particularly problematic considering there is significant toxicity, hate, bias, and negativity present in the large-scale web crawled datasets text generation models are commonly trained on. Imposing conditions or constraints on the generation process results in safer and more useful generated texts for downstream application tasks (Krause et al., 2021).

## 2.2 Conditional Text Generation

Conditional text generation manipulates attributes of the generated content depending on specific contexts or user needs, and it allows the data generation process to focus on specific modes of the data. Conditioning the generative model on additional information makes it possible to generate texts which satisfy given *input* conditions and meet desired attributes. In the literature conditional text generation is sometimes referred to as context-dependent text generation. While the word context may carry different semantics for different readers, in this survey we consider as context only attributes which are inherently external to the model itself; model intrinsic attributes such as for example, the history of past generated words, is already included in the formulation of generic text generation. For example, context attributes used for conditioning generated texts are the *source sentence* in machine translation, the *conversational history* in dialogue systems, the *input document* in text summarization and text simplification, the *input question* in question answering systems, the *prompt* given to a large language model, or contextual information such as *product, time*, and *location* in review generation.Essentially, conditional text generation is a form of sequence-to-sequence generation, given the input condition is a text sequence and the goal is to generate another output sequence.

Conditional text generation models add a contextual variable or attribute code $c$ to the probabilistic model $p(\boldsymbol{x})$ transforming it into a conditional probability model $p(\boldsymbol{x}|c)$, which can be auto-regressively decomposed using the chain rule of probability $p(\boldsymbol{x}|c) = \prod_{i=1}^{n} p(x_i|\boldsymbol{x}_{<i}, c)$. When $p(\boldsymbol{x}|c)$ is modeled by a neural network with parameters $\theta$, the model minimizes the negative log-likelihood loss function accounting for the attribute code $c$: $\mathcal{L}(D) = -\sum_{k=1}^{|D|} \sum_{i=1}^{N_k} \log p_\theta(x_i^k|\boldsymbol{x}_{<i}^k, c^k)$. Besides generation, conditional models can also be used as generative classifiers to compute $p(c|x_{<i})$ by applying Bayes rule.

## 2.3 Constrained Text Generation

The problem of constrained text generation is focusing on generating coherent and logical texts that do (not) cover lexical concepts (for eg., pre-defined nouns, verbs, entities, phrases or sentence fragments) desired to be (not) present in the *output*, as well as generate outputs that abide to specific format, semantic, syntactic or utility rules to reflect the particular interests of the system user. Constraints impose restrictions on the generative model that must be satisfied by any solution to the optimization problem and their fulfillment can be tested accordingly. In the literature the distinction between conditional, controlled, and constrained text generation is not clearly defined, and these terms are often used interchangeably. In fact, the first work that proposed generating constrained text is actually referring to the task as "controlled" generation (Hu et al., 2017). In what follows we formally define the problem of constrained text generation.

Let us consider we are (optionally) given an unordered or ordered set of $n$ concepts $c = \{c_1, c_2, \ldots, c_n\} \in \mathcal{C}$, where $\mathcal{C}$ denotes the space of all concepts, and $c_i \in C$ is a concept belonging to the concept vocabulary. In addition, let us assume we are also (optionally) given a set of $m$ rules $z = \{z_1, z_2, \ldots, z_m\} \in \mathcal{Z}$, with $z_i \in \mathcal{R}$, where $\mathcal{R}$ denotes the space of all rules, and each $z_i$ is a text generation constraint expressed in logical form. We formulate constrained text generation as learning the structured predictive function $f : \mathcal{C} \cup \mathcal{Z} \rightarrow \mathcal{X}$, where $\mathcal{C} \cup \mathcal{Z} \neq \phi$ which maps a set of concepts and/or constraint rules to a generated sentence. Therefore, constrained text generation methods impose constraints on the generated sentences and produce output in the form of grammatical sentence $y \in \mathcal{Y}$ which contains all concepts present in $c$ and all constraint rules specified in $z$. The probability $p(y|f)$ can still be modeled autoregressively $p(y|f) = \prod_{i=1}^{n} p(y_i|\boldsymbol{y}_{<i}, f)$; when $p(y|f)$ is modeled by a neural network with parameters $\theta$, the negative log likelihood function can be minimized while leveraging $f$ for constraint satisfaction $\mathcal{L}(D) = -\sum_{k=1}^{|D|} \sum_{i=1}^{N_k} \log p_\theta(y_i^k|\boldsymbol{y}_{<i}^k, f)$.

The matching function $f$ manipulates the probability distribution and indicates to which extent the constraints are satisfied. In the literature, constrained text generation methods can be either *i) Soft-constrained (priming)*, when the matching function $f$ is a soft measure of semantic similarity and only requires the generated sentences to be semantically related to the given constraints, or *ii) Hard-constrained*, when the matching function $f$ is a binary indicator which rules out the possibility of generating infeasible sentences that do not meet the given constraints. Hard-constrained text generation is notably a more challenging task compared to soft-constrained text generation, and it requires specialized approaches and architectures to ensure the constraints in the output sentence. In contrast, soft-constrained text generation models are usually easier to design, e.g., with the use of existing copy and attention mechanisms for soft enforcing constraints and annotated keyword-text pairs; nevertheless, even soft constraints are likely to be lost during generation, especially if multiple weakly correlated (lexical) constraints must be included (Zhang et al., 2020b).

Compared to generic text generation which assumes no conditions on input or output other than existing context, and compared to conditional text generation which places conditions on the input which can be considered at training time, constrained text generation places conditions on the output which is a considerably more difficult and challenging problem to solve. Unlike input conditions, output conditions cannot be considered at training time and their satisfaction is assessed after training has completed by sampling and inspecting the generated outputs. In addition, standard sequence generation architectures are not designed to easily accommodate or incorporate output constraints. Given the model structure itself cannot express output conditions, it becomes challenging to evaluate the extent to which constraints are satisfied by a model, objectively compare and contrast the performance of different models, and measure overall success to inform on progress in constrained natural language generation. Due to these limitations, current methods proposed to address constrained text generation are neither satisfactory nor sufficient. The main machine learning challenge is that it is hard to evaluate the objective function for constrained text generation, and very few studies have approached the problem from the prism of editing the objective function to incorporate constraints at training time. Even if constraints were to be added to the objective function itself, constrained optimization would be another challenge. In general, reinforcement learning approaches are used in the context of text generation to optimize non-differentiable reward functions computed at the token level, for eg., BLEU in machine translation or ROUGE in text summarization. However, optimizing such automatic measures that focus on local n-gram patterns often results in deteriorated textual outputs despite increased automatic scores (Bosselut et al., 2018; Pasunuru et al., 2020). Moreover, applying reinforcement learning to text generation at the word level leads to difficulty in proper temporal credit assignment for long-term textual rewards (Saleh et al., 2020). Given that the environment provides only delayed rewards as the agent executes a sequence of actions, it is often impossible to know whether the agent succeeds in achieving a task until the end of the episode, at which point the agent needs to determine which of the actions in the sequence are to be credited with producing the resulting reward (Gao et al., 2019a). Adding constraints on top of existing reinforcement learning issues would be detrimental to the learning process, if not make learning close to impossible: the objective function would be even harder to optimize, rewards would be delayed, sparse and non-informative. Despite these open problems and limitations, we argue neural constrained text generation is an important research area which deserves a lot more attention.

Constrained text generation is useful in many scenarios, such as incorporating in-domain terminology in machine translation (Post & Vilar, 2018), improving semantic corectness (Balakrishnan et al., 2019), avoiding

generic and meaningless responses in dialogue systems using grounding facts (Mou et al., 2016), paraphrase generation in monolingual text rewriting (Hu et al., 2019; Kajiwara, 2019), incorporating ground-truth text fragments (such as semantic attributes, object annotations) in image caption generation (Anderson et al., 2017), creating a story (Fan et al., 2018b) or poem (Ghazvininejad et al., 2017) using a pre-defined set of keywords, or re-writing a user search query as a fluent sentence. Typical attributes used to generate constrained natural language are the *tense* and the *length* of the summaries in text summarization (Fan et al., 2018a), the *sentiment* of the generated content in review generation (Mueller et al., 2017), *language complexity* in text simplification or the *style* in text style transfer applications. In addition, constrained text generation is used to overcome limitations of neural text generation models for dialogue such as genericness and repetitiveness of responses (See et al., 2019; Serban et al., 2016).

Nevertheless, generating text under specific lexical constraints is challenging. Common models and architectures employed for natural language generation are autoregressive in nature, generating tokens one by one in a sequential manner from left to right; by design, these models lack fine control over the generated sequence and cannot easily support constraints at arbitrary positions in the output or constraints involving multiple input objects (Zhang et al., 2020b; Hsieh et al., 2021). While for humans it is straightforward to generate sentences that cover a given set of concepts or abide to pre-defined rules by making use of their commonsense reasoning ability, generative commonsense reasoning with a constrained text generation task is more challenging for machine learning models (Lin et al., 2019).

## 3 NLG Constraints

Natural language generation models place restrictions on the generated output to produce texts that reflect certain user preferences. In Table 1 we present NLG tasks distinguishing between conditions and constraints. We broadly group existing constraints into the following categories:

**Lexical constraints** Lexical constraints allow to include specific keywords, phrases or entities at arbitrary positions in the output, and can be specified as a word (a single token) or phrasal constraint (a multi-word phrase). They are useful in tasks such as *dialogue/poetry generation*, *machine translation*, *story telling*, etc.

**Format constraints** Format constraints such as number of sentences, length of sentences, order of words, number of syllables, etc. serve to denote preferences on the form and appearance of the generated output. Format constraints are particularly useful in tasks such as *poetry generation* to specify the form of the generated poem, for example quatrain or regulated verse, length of the poem, rhyme and rhythm. In *text summarization* or *text simplification*, length constraints define the length of the generated output to be strictly less than the length of the input document, while in *dialogue generation* they help define the level of verbosity of the dialogue agent.

**Semantic constraints** Semantic constraints are used to define the topic and sentiment of the generated content, or control fine-grained aspects such as removing toxicity. Topic constraints are particularly useful in *dialogue generation*, where the goal is to generate on-topic responses that are safe, non-harmful, unbiased, relevant to the dialogue context and particular user needs; in *story telling* or *poetry generation*, topic constraints help define the theme. Generating language that conveys particular positive, neutral or negative sentiment aims to endow artificial agents with human-like traits such as compassion, empathy, and enables agents to react with appropriate emotion in diverse social situations; constraining on a specific sentiment is important in many tasks such as *dialogue generation*, *review generation*, *story telling*, *poetry generation* or *text style transfer*. Furthermore, increasing politeness of a dialogue system or reducing toxicity of generated language are important aspects with respect to human-centered metrics of conversation quality.

**Syntactic constraints** Syntactically constrained text generation produces sentences with desired syntax by incorporating syntactic templates and rules in the training of the text generative model. Syntactic constraints are useful in paraphrase generation, where given a sentence and a target syntactic form (e.g., a constituency parse), a system must produce a paraphrase of the sentence whose syntax conforms to the target (Iyyer et al., 2018). Generating texts that convey the same meaning but with different expressions has

Table 1: Overview of constrained NLG tasks, differentiating between conditions and constraints.

| Task | Condition | Constraint | | | | |
|---|---|---|---|---|---|---|
| | | *Lexical* | *Format* | *Semantic* | *Syntactic* | *Utility* |
| **Machine Translation** | source input | words phrases entities | – | topic sentiment | paraphrase tense gender pronouns | target language politeness factuality/faithfulness helpfulness, harmlessness |
| **Dialogue Generation** | past utterance(s) | words phrases entities | length verbosity | topic sentiment toxicity | paraphrase gender pronouns | politeness personality traits factuality/faithfulness helpfulness, harmlessness |
| **Text Summarization** | input document(s) | words phrases entities | length | topic | paraphrase | factuality/faithfulness helpfulness, harmlessness |
| **Text Simplification** | input text | words phrases entities | length | topic | paraphrase | simpler vocabulary readability factuality/faithfulness helpfulness, harmlessness |
| **Text Style Transfer** | source text | words phrases entities | length | topic sentiment | paraphrase tense gender pronouns | style factuality/faithfulness |
| **Question Answering** | input question | words phrases entities | length | topic | paraphrase tense gender pronouns | factuality/faithfulness politeness helpfulness, harmlessness |
| **Narrative Generation/ Story telling** | – | words phrases entities entities | length | topic sentiment | paraphrase tense gender pronouns | readability factuality/faithfulness helpfulness, harmlessness style |
| **Poetry Generation** | – | words phrases entities | length rhyme rhythm | topic sentiment | paraphrase tense gender pronouns | readability factuality/faithfulness style |

numerous applications in many natural language generation tasks, including monolingual transduction tasks such as *text simplification*, *text compression*, or *text style transfer*, as well as in tasks like *text summarization*, *machine translation* or *question answering* where alternative ways of expressing the same information capture the inherent language variations.

**Utility constraints**  Utility constraints capture holistic properties of the generated output, for example, stylistic, readability, faithfulness, helpfulness, harmlessness or politeness aspects. Preserving the information content of texts while manipulating attributes such as style, readability level, personality traits of the user or specific gender pronouns allows to customize generated texts to different audiences and make them relevant in a wide variety of end-user applications. Stylistic constraints are immediately relevant to the task of *text style transfer*, with applicability in many tasks, including *dialogue generation*, *machine translation*, *text simplification*, *story telling*, *poetry generation*, *review generation*.

Constraining text generation on attributes such as readability and level of text complexity serves to adapt the generated output to users of different age, backgrounds and educational levels. Reducing complexity of texts while preserving the information content is the main goal of *text simplification*; in addition, in tasks such as *dialogue generation*, *text summarization*, *story telling*, *poetry generation*, *question answering* it is important to customize texts for various literacy levels.

In many languages the degree of politeness is an important aspect of inter-personal communication, and honorifics are used to express courtesy, social distance, or the relative social status between the speaker and their addressee(s) (Sennrich et al., 2016). Politeness constraints on the output are used in *machine translation*, *dialogue generation*, *story telling*, and *text style transfer*.

Faithfulness constraints enforce similarity between a generated text sequence and its corresponding input, requiring models to generate texts that are faithful, factual and preserve the original information content. Such constraints are important in many tasks, including *text summarization*, *machine translation*, *text simplification* or *dialogue generation*, where models are vulnerable to producing hallucinated content.

Language constraints are useful when translating texts between different languages such as in *machine translation*, or from complex language into simple language such as in *text simplification*.

## 4 Constrained Natural Language Tasks

In what follows we briefly describe major NLG tasks and differentiate between the roles of conditions and constraints in these tasks.

**Machine Translation**  Machine translation is focusing on the automatic translation of textual content from one language into another language, and is a typical example of both *conditional and constrained text generation*, as it conditions on the input text in the source language and constraints the model to generate fluent and faithful output in the target language. Additionally, constraints can be placed on the degree of formality and politeness, the use of gender-specific pronouns, the inclusion in the target sentence of named entities or specific concepts from the source sentence.

**Dialogue Systems**  A dialogue system, also known as a conversational agent, is a computer system designed to converse with humans using natural language. Dialogue generation is an instance of *conditional text generation* where the system response is conditioned on the previous user utterance and frequently on the overall conversational context (e.g., a prompt given to Large Language Model based Chatbots). Dialogue generation can also be an instance of *constrained text generation* - it is desirable that generated dialogues incorporate explicit personality traits (Zheng et al., 2019), control the sentiment (Kong et al., 2019), topic, degree of formality and politeness of the generated response to resemble human-to-human conversations. In addition, dialogue responses may need to incorporate text excerpts from past dialogue history or entities such as locations, persons, institutions, etc. From an application point of view, dialogue systems can be categorized into: *i) task-oriented dialogue agents*, designed to help users complete a particular task, or *ii) non-task oriented dialogue agents (chat-bots)* designed to carry entertaining conversations with their users on a wide range of open domains. A common problem in dialogue generation systems is that they tend to generate safe, universally relevant responses that carry little meaning (Serban et al., 2016; Li et al., 2016a; Mou et al., 2016). Moreover, they can fail to take turns asking questions and balance specificity with genericness of the output (See et al., 2019).

**Text Summarization**  Text summarization facilitates a quick grasp of the essence of a document and produces a condensed version of its content, by copy-pasting the relevant portions from the input as in extractive summarization (Nallapati et al., 2017), or by generating novel content as in abstractive summarization (Rush et al., 2015; Nallapati et al., 2016; See et al., 2017), or via hybrid approaches (Liu et al., 2018) that combine both techniques. Text summarization is a *conditional text generation task* where the condition is represented by the given document(s); additional conditions are used in remainder summarization to flexibly define which parts of the document(s) are of interest, for eg., remaining paragraphs the user has not read yet, or in source-specific summarization to condition summaries on the specific input source and style of writing, for eg., newspapers, books or news articles. Text summarization is also a *constrained text generation* task considering that the length of the summary is fixed, pre-determined, and strictly less than the original document. The goal of text summarization is to allow the user to digest information at different levels of granularity and detail according to personal needs, interests and time budget. Moreover, constraints can be placed on specific concepts to include in the summary, such as named entities, or on explicitly picking sentences from the original document as in extractive summarization.

**Text Simplification**   Text simplification is designed to reduce the text complexity, while preserving its original meaning. In the literature, simplification has been addressed at multiple levels: *i) lexical simplification* focused on replacing complex words or phrases with simpler alternatives; *ii) syntactic simplification* alters the syntactic structure of the sentence; *iii) semantic simplification* paraphrases portions of the text into simpler and clearer variants. End-to-end models attempt to combine all these steps. Text simplification is both *conditional and constrained text generation*; we are conditioning on the input complex text to generate a simpler version, accounting for constraints such as higher readability, simpler vocabulary, and shorter sentence length than the complex input.

**Text Style Transfer**   Style transfer has its origins in computer vision applications for image-to-image translation and more recently has been used in natural language processing applications for machine translation, sentiment modification to change the sentiment of a sentence from positive to negative and vice versa, word substitution decipherment and word order recovery (Hu et al., 2017). Text style transfer is designed to preserve the information content of a source sentence while altering the way it is delivered to meet desired presentation constraints. Textual content is disentangled from the style in which it is presented, and manipulating stylistic attributes can be done without parallel aligned data between source and target styles. Text style transfer is an instance of both *conditional and constrained text generation* given that we condition on the given source text and constrain the transferred sentences to stylistically match target examples.

**Question Answering**   Question answering systems are designed to find and integrate information from various sources to provide responses to user questions (Fu & Feng, 2018). While traditionally candidate answers consist of words, phrases or sentence snippets retrieved and ranked appropriately from knowledge bases and textual documents (Kratzwald et al., 2019), answer generation aims to produce more natural answers by using neural models to generate the answer sentence. Question answering is both *conditional and constrained text generation* task; the system conditions on the user question, and simultaneously ensures that concepts needed to answer the question are present in the generated output. Diverse question answering systems are proposed in the literature addressing for eg., medical information needs (Wiese et al., 2017), mathematical questions (Schubotz et al., 2018), quiz bowl questions (Iyyer et al., 2014), cross-lingual and multi-lingual questions (Loginova et al., 2018). Notably, in practical applications users are not only interested in learning the exact answer word or phrase, but also in how it relates to background information and to previously asked questions and answers (Fu & Feng, 2018).

**Narrative Generation/Story Telling**   Neural narrative generation is an important step towards computational creativity (Gervás, 2009) and represents a long-form open-ended text generation task which simultaneously addresses the selection of appropriate content (*"what to say"*) and the surface realization of the generation (*"how to say it"*)(Wiseman et al., 2017). Narrative generation is a *constrained text generation* task that places explicit constraints on concepts to steer the narrative in particular topic directions and expands the few keywords specified as the story title, beginning or ending. While existing models can generate stories with good local coherence, generating long stories is challenging. Difficulties in coalescing individual phrases into coherent plots and in maintaining character consistency throughout the story lead to a rapid decrease in coherence as the output length increases (van Stegeren & Theune, 2019). Hierarchical models for story generation break down the generation process into multiple steps: first modelling the action sequence, then the story narrative, and finally entities such as story characters (Fan et al., 2019). Neural narrative generation combining story-writing with human collaboration in an interactive way improves both story quality and human engagement (Goldfarb-Tarrant et al., 2019).

**Poetry Generation**   The poem generator operates in an interactive context where the user supplies the model with a set of ordered concepts that reflect her writing intent, as well as the format of the poem, for eg. quatrain or regulated verse. Poetry generation is a *constrained text generation* problem since user defined concepts need to be included in the generated poem, and a *conditional text generation* problem given the explicit conditioning on stylistic attributes. For a detailed overview of poetry generation see (Oliveira, 2017).

# 5 Constrained NLG Methods

Accounting for the different types of constraints introduced in Section 3, we distinguish the following methodologies commonly employed in the constrained text generation literature: *i)* decoding approaches, *ii)* fine-tuning approaches, *iii)* discriminative approaches, *iv)* edit-based approaches, *v)* adapting existing models and architectures to accommodate constraints on the generated output, and *vi)* prompting large language models. In what follows we present each approach in detail, outlining the main associated challenges.

## 5.1 Decoding approaches

The most popular approach to text generation in the literature has been supervised learning with task-specific datasets; nevertheless since many real-world applications require diverse and potentially evolving constraints, it is infeasible to annotate task-specific training data for every combination of constraints (Qin et al., 2022). Even if collecting the data was not a bottleneck, re-training large language models that are extreme in scale for each new constraint or combination of constraints is undesirable. The alternative to fine-tuning language models with task-specific datasets is to enrich decoding algorithms so as to accommodate constraints on the fly. We present decoding approaches to constrained text generation below.

**Lexical constraints** *Lexically constrained (guided) decoding* aims to restrict the search space at decoding time to sequences which contain pre-defined lexical constraints only. These lexical constraints can be specified in the form of a word constraint (a single token) or a phrasal constraint (a multi-word phrase, i.e. a sequence of two or more contiguous tokens). To this end, the beam search decoding algorithm is modified to enforce the inclusion of pre-specified words and phrases in the generated output by allowing the model distribution to not only account for the given lexical constraints, but also to generate parts of the output sequence not covered by the constraints. In general, the decoder can more easily place multiple sequential tokens in a phrasal constraint (where the permutation order is fixed) on the generated output as opposed to placing multiple separate, independent constraints. In addition, the lexically constrained decoding approach assumes lexical constraints are pre-determined, which may not always be the case; if so, the open question is where to get lexical constraints from.

Early work on constrained decoding in machine translation relies on the *placeholder approach* designed to recognize identifiable elements (numbers and named entities) in the source sentence, temporarily replace these with corresponding placeholders during preprocessing, and then substitute the assigned placeholders with the original source-language strings during beam search decoding (Crego et al., 2016; Iso, 2024). Nevertheless, such an approach is limited and unable to model the source tokens in target language specific terminology or the vocabulary from a new out-of-distribution domain. *Prefix decoding* represents a modification of beam search to first ensure that a user defined target prefix is generated first, and only after build hypotheses for the suffix that maximize the coverage of the remaining source-side tokens. As decoding progresses from left to right, the decoder transitions from a constrained prefix decoding mode to unconstrained beam search. For example, the start of the sentence symbol $$ can be easily included as the first word of a constraint (Knowles & Koehn, 2016; Wuebker et al., 2016). In the context of text summarization, an essential property of a summarization system is the ability to generate a summary with desired length. *Grid beam search* (Hokamp & Liu, 2017) extends beam search decoding to allow for the inclusion of arbitrary target side hard lexical constraints at any position in the generated sequence. Given $C$ input constraints, the algorithm maintains $C+1$ separate beams $B_0, B_1, \ldots, B_c$ that group together hypotheses which meet the same number of satisfied constraints. Decoding runs similar to beam search, with an additional dimension added to keep track of how many constraints are met by each hypothesis at every timestep; the highest scoring hypothesis in beam $B_c$ is ultimately generated. However, grid beam search is impractical as decoding complexity is linear in the number of constraints, i.e. beam size increases proportionally to the amount of constraints and changes for every sentence. *Constrained beam search* (Anderson et al., 2017) guarantees the inclusion of input constraints in the generated sentences by extending beam search with a finite state machine whose states mark completed subsets of the input set of constraints; however, decoding complexity has an exponential cost in the number of constraints, making it infeasible in many applications. *Dynamic beam allocation* (Post & Vilar, 2018) improves upon the runtime complexity of grid beam search and constrained beam search by decoding with constant complexity $O(1)$ in the number of constraints. The algorithm still groups

together hypotheses that have met the same number of constraints by using a single fixed-size beam which is dynamically divided at each time-step according to how many constraints have been met. Despite being more efficient, dynamic beam allocation does not necessarily outperform conventional beam search (Lin et al., 2019). In addition, the generation of hypotheses that only partially satisfy a phrasal constraint needs to be aborted to unwind to the tokens in the constraint. **Neurologic decoding** (Lu et al., 2021) modifies beam search to enforce the satisfaction of lexical constraints expressed under predicate logic in conjunctive normal form (CNF). Given the intractability of exhaustive beam search to optimize CNF constraints, the algorithm searches for approximately-optimal output sequences in which all clauses are satisfied, including both positive and negative constraints (i.e. words that must be generated, respectively omitted in the output sequence). The method is applied to cooking recipe generation, where the task is to generate cooking instructions given a dish name and a list of ingredients, and to data-grounded dialogue response generation where a response is generated given a query and a list of facts to convey.

In general, lexically constrained decoding methods have high computational complexity and force the inclusion of specific words in the generated sentence at every timestep of the generation process with no prior examination of these specific words before generation begins (Latif et al., 2020); this unnatural way of generating sentences can impact the quality and naturalness of the generated output (Liu et al., 2019a; Post & Vilar, 2018). In addition, there is a trade-off between the generated text quality and hard constraint satisfaction (Iso, 2024). In a lack of suitable evaluation metrics, there is no commonly agreed criteria for objectively assessing the quality of the generated sentences and conducting comparisons across text generation models.

**Format constraints** *Fixed length decoding* (Kikuchi et al., 2016) constrains the length of generated summaries in two ways: *i)* by preventing the decoder from generating the end-of-sentence tag until the length of the generated sequence exceeds the desired length, and *ii)* by defining the minimum and maximum length range of the sequence and discarding out-of-range sequences. **Non-monotonic decoding** approaches allow tokens to be inserted at any position in the generated sequence during decoding, therefore accommodating flexible orderings of the output. Unlike left-to-right autoregressive generation that produces a single word at a time, non-monotonic decoding can satisfy lexical constraints at multiple locations in the output sequence allowing for highly parallel generation and faster decoding times. Nevertheless, such approaches assume the generated sequence length is known a priori, preventing it from being dynamically adjusted as generation proceeds. Moreover, such models assume conditional independence between output tokens, i.e. tokens are generated independently, may be inconsistent and agnostic to each other. Consequently, this approach may hurt the expressiveness of the model and lead to potential performance degradation, impacting the fluency and naturalness of the output. In addition, non-monotonic sequence decoding approaches can terminate prematurely before constraints are satisfied in the output sequence (Zhang et al., 2020b; Hsieh et al., 2021). The main limitation of this approach is the lack of model expressiveness in accommodating constraints.

Insertion Transformer (Stern et al., 2019) proposes a flexible sequence generation framework based on repeated insertion operations into an initially empty output sequence until a termination condition is met. The model adopts a progressive masking approach based on token importance in the original text and is trained to generate a missing token between every two tokens in the input. To this end, the original Transformer (Vaswani et al., 2017) decoder is modified to allow insertions not just at the end but anywhere in the output sequence. The model can decode sequences serially one token at a time, or it can decode sequences in parallel with simultaneous insertions at multiple locations. A similar approach is considered in InDIGO (Gu et al., 2019a) which extends Transformer for insertion-based decoding with inferred generation order. Token generation order for the output sequence is modeled as a latent variable, and at each decoding step the model predicts both the generated word and its position in the output sequence; nevertheless, strong conditional independence is assumed between the output tokens which hurts output quality. An iterative refinement step based on latent variables is added to the Transformer decoder to refine a target sequence gradually over multiple steps until a predefined stopping criterion is met (Lee et al., 2018). Progressive Insertion Transformer (Zhang et al., 2020b) uses non-autoregressive modeling based on a top-down progressive structure for lexical hard-constrained text generation. Given lexical constraints as input, the model inserts tokens progressively according to word importance to generate the target sequence, as follows: first it generates high-level words in a sentence such as nouns, adjectives and verbs, then uses these as pivoting points to insert details of finer granularity and finally completes the sentence by adding connecting words which

carry less information, such as pronouns and prepositions. Entity Constrained Insertion Transformer (Hsieh et al., 2021) builds upon previous models considering hard lexical constraints in the form of entities in the output sequence. Similar approaches train the Transformer decoder to insert missing tokens in a partially complete sequence without relying on a pre-specified factorization of tokens (Chan et al., 2019; Gu et al., 2019b); based on the information available in the sequence, the insertion-based generative model is able to dynamically infer the remaining parts irrespective of their arbitrary order.

**Syntactic and Semantic constraints** *Distributional constraints* (Baheti et al., 2018) on topic and semantic similarity are used to incorporate source side-information at decoding time in neural conversational systems and encourage the generation of more diverse responses. Moreover, constraints over topics and syntax are used to generate matching or semantically similar statements in response to the user input (Niu & Bansal, 2018). Lexically constrained decoding from pre-trained language models aims to steer language models in useful and safe directions so as to minimize the risks associated with these models generating biased, offensive and toxic content (Sheng et al., 2019; Holtzman et al., 2020). **Weighted decoding** methods employ a linear combination of output logits from multiple prompts (raw prompt vs prefix-prepended prompt) to vary the strength of desired target attributes in the output text (Pei et al., 2023; Zhang & Song, 2022). **Energy-based constrained decoding** allows the specification of style and lexical constraints through an energy function and performs differentiable reasoning through gradient-based sampling (Qin et al., 2022), (Mireshghallah et al., 2022). Constrained text generation is viewed as an optimization problem, where the goal is to iteratively seek text with lower energy. The sampling process uses gradients of the energy function to update a continuous relaxation of text data, which is then mapped back to the discrete space of natural language via a discretization approach. For streering the generation towards desired constraints, biases are applied to the logits of the pre-trained model output layer, which is also found to improve the speed of the decoding process (Liu et al., 2023b). Nevertheless, sampling from energy-based models requires many iterations to converge to plausible text.

## 5.2 Fine-tuning approaches

**Semantic and Utility constraints** Controlling the output of pre-trained language models is crucial in a wide-range of safety-critical applications, including mental health support chatbots, sentiment controlled text generation, language detoxification, etc. To this end, fine-tuning approaches are used for fine-grained control over individual stylistic aspects (for eg., length, professional and descriptive style, tense, personal voice, gender) and content aspects (for eg., sentiment and topic) of the generated texts (Ficler & Goldberg, 2017), (Lample et al., 2018). Typically, the pre-trained model is fine-tuned separately for each attribute of interest, which poses the challenge of how to learn disentangled latent representations of style and content in neural language models (John et al., 2019) and isolate the desired attribute from the distribution shift between the generative model and the fine-tuned dataset. The lack of datasets that are diverse and representative of constrained criteria encountered in practice represents an open challenge for fine-tuning pre-trained models.

CTRL (Keskar et al., 2019) uses control codes to generate texts that meet user-defined constraints on domain, style, topics, dates, entities, relationships between entities, plot points, and task-related behavior. These pre-defined codes are appended at the beginning of raw text sequences to define task-specific training data and create controllable task-specific behaviour at sampling time. Similarly, fine-grained semantic control codes are used to steer generation towards targeted attributes (Ross et al., 2022). Decoding Experts (DExperts) (Liu et al., 2021a) is a decoding-time method for constrained text generation which combines a pre-trained language model with both an "expert" and "anti-expert" language model in a product of experts. The "expert" models desirable aspects of the generated text (for eg., positive sentiment), while the "anti-expert" plays the antagonistic role of modeling undesirable attributes to be avoided (for eg., toxicity); each one of the three language models is conditioned on the same user prompt. While the method highlights the promise of customizing decoding from pre-trained language models in safe and efficient ways, gathering large amounts of toxic data to model undesirable attributes may be challenging. In general, adding negativity to a positive prompt is a much easier task than adding a positive turn to a negative prompt (Madotto et al., 2020). Augmenting the set of positive examples commonly used for LLM training with a set of negative examples, i.e. completions given a prompt that a model should not generate, helps reduce the likelihood of repetitive model generations and negative tokens occuring in the output (Adolphs et al., 2023). Similarly,

token-level or sequence-level objectives are used to discourage LLM models from assigning high probabilities to certain tokens or sequences (Lu et al., 2022; Liu et al., 2021b). Aligning LLM models with user preferences at inference time is important for personalizing LLM models to their users without the need to retrain the model for each new target attribute. SteerLLM (Dong et al., 2023) allows end-users to customize responses during inference by conditioning a supervised finetuned model on a multi-dimensional set of user attributes.

Reinforcement learning from human feedback (RLHF) (Christiano et al., 2017; Stiennon et al., 2020) is a key component in improving the instruction-following and generation abilities of LLM models. Aligning AI models with human preferences is considered crucial for safely deploying artificial systems in the real-world and ensuring they exhibit behaviors consistent with human values (Ouyang et al., 2022; OpenAI, 2022; Ziegler et al., 2019; Shen et al., 2023; Wang et al., 2024b). RLHF algorithms further finetune under a KL-constrained RL objective a language model that has already undergone supervised fine-tuning (referred to as the reference model); this objective encourages the model to maximize the reward and simultaneously discourages high KL divergence between the language model and the reference model. The reward model is derived from human preferences on text continuations with positive sentiment or vividly descriptive language, while the KL constraint is used to prevent the fine-tuned policy from drifting too far from the reference policy. In dialogue systems, KL control has been used to retain prior information and penalize divergence from the pre-trained model during RL fine-tuning (Jaques et al., 2019). Despite the promise of RLHF in aligning LLMs to user preferences, fine-grained control over large language models remains a significant challenge (Wang et al., 2024a). Since RLHF training can be difficult and unstable, alternative approaches finetune language models to mimic the Best-of-N (BoN) (Nakano et al., 2021; Touvron et al., 2023) distribution by minimizing the KL divergence between the language model and the BoN distribution, for example when generating movie reviews with positive sentiment (Amini et al., 2024) or in text summarization (Gui et al., 2024). Nevertheless, KL does not capture attributes of generated text that humans judge to be salient, such as the length of the generated response (Singhal et al., 2024). The lack of suitable evaluation metrics that correlate with human judgements is a bottleneck in generating high quality outputs.

Pre-trained OpenAI-GPT2 (Radford et al., 2019) model is used to re-write a story through counterfactual reasoning and generate a narrative consistent with the imposed constraints (Qin et al., 2019). In abstractive summarization, OpenAI-GPT2 is used in a reinforcement learning setting which trains the summarization agent to maximize coverage and fluency of the generated content constrained on a pre-defined length (Laban et al., 2020). RecipeGPT (H. Lee et al., 2020) fine-tunes the GPT-2 pre-trained language model for generating cooking instructions when hard constraints are placed on the recipe title and ingredients; the model can also generate the list of ingredients for a recipe when constrained on the recipe title and specific cooking instructions. Infilling by language modeling is used to complete variable length text spans (e.g. words, n-grams and sentences) by fine-tuning a pre-trained language model on sentence pairs that contain both artificially-masked text and the corresponding original text (Donahue et al., 2020).

While fine-tuning models on task specific datasets has become the dominant paradigm for constrained text generation from pre-trained large language models, these models generally fail to reliably incorporate the underlying constraints in the generated texts even when supervised with large amounts of task-specific examples (Lu et al., 2021). Moreover, the superficial alignment hypothesis (Zhou et al., 2024) argues that that almost all knowledge in large language models is learned during pre-training and that alignment tuning only teaches the base LLM in which data format and language style to interact with its users. Further analysis of this hypothesis finds that fine-tuning approaches only influence a small number of tokens focused primarily on stylistic elements (Lin et al., 2024a). The lack of model expressivity to incorporate constraints in an important challenge for fine-grained constrained text generation.

**Format constraints** Due to length biases in reward models, LLM models fine-tuned with RLHF tend to suffer from verbosity issues and generate long answers that are not necessarily of high quality (Singhal et al., 2024; Park et al., 2024b; Dubois et al., 2024b; Kabir et al., 2023; Nakano et al., 2021; Sun et al., 2023b; Wu et al., 2024a). Offline preference optimization algorithms with implicit reward are known to exploit evaluator length biases and quickly increase the length of the generated text during training without capturing more complex features of human preferences (Rafailov et al., 2024; Li et al., 2023b; Dubois et al., 2024a). To disentangle verbosity from quality, length-controlled direct preference alignment methods employ

an additional regularization term in the loss function that governs the token length of the generated response (Park et al., 2024b). Online RLHF approaches that first train a reward model include a similar length regularization term in the reward modeling stage (Chen et al., 2024a). Open problems with current RLHF frameworks include their limited ability to capture and adapt to the complexity of user-dependent preferences in the real world (Wang et al., 2024a). Since human preferences can change over time depending on users and their expectations, methods relying on multi-objective reward models are used to capture desirable aspects of the generated texts (eg., verbosity, factuality, helpfulness, harmlessness) (Pan et al., 2023; Rame et al., 2024a; Dong et al., 2023; Bakker et al., 2022; Wu et al., 2024a). To enhance the control and personalization of a single LLM model across desired range of attributes, different user preferences are encoded as unit vectors and embedded numerically into the system prompt (Wang et al., 2024a; Yang et al., 2024b; Dong et al., 2023). The analysis of the alignment tuning process reveals that RLHF simply teaches the base LLM model to select a sub-distribution of data formats for interacting with the user (Lin et al., 2024a). Notably, most distribution shifts occur with stylistic tokens (transitional phrases, discourse markers, safety disclaimers) instead of content-bearing words, which supports the *superficial alignment hypothesis* (Zhou et al., 2024).

### 5.3 Discriminative approaches

**Utility constraints**  One of the early works proposing constrained text generation learns disentangled latent representations by combining variational auto-encoders with attribute discriminators (Hu et al., 2017). Semantic structure is imposed on the latent codes by using global discriminators, one for each attribute, to guide the learning of the discrete text generator and force it to allocate one latent dimension per attribute code. The model is used to manipulate the sentiment and tense of the generated sentences.

Weighted decoding (Holtzman et al., 2018) relies on a mixture of discriminative models to guide a recurrent generator towards incorporating attributes that enhance the overall coherence, style, and information content of the generated text. The discriminators complement each other and their weighted contributions form the final decoding objective from the generator. Similarly, stylistic configurations are revised and polished for generated poems by adding additional weights during decoding to control the style of generated poem, including the repetition, alliteration, word length, cursing, sentiment, and concreteness (Ghazvininejad et al., 2017). Nevertheless, modifying the scoring function used for generation as in weighted decoding often leads to sacrificing fluency and coherence of the generated text (See et al., 2019). Selective sampling (Wang et al., 2017) relies on a sample selector (multilayer perceptron for binary classification) which outputs whether the current sample should be accepted or rejected based on the presence of desired target words that define the output style and topic in the generated sequence. The robustness of evaluation metrics is directly correlated with model performance, therefore it is crucial to focus on developing metrics that capture diverse aspects of text quality during training and sampling time.

Generating texts with desirable attributes from a pre-trained unconditional language model $P(X)$ is a non-trivial task. Most approaches resort to either training from scratch a new conditional model $P(X|a)$ for desired attribute $a$, or fine-tuning $P(X)$ on additional data representative for the attribute $a$. Theoretically, rejection sampling could also be used to sample $P(X|a)$ from $P(x)$, but this approach is highly inefficient in practice. Fudge (Yang & Klein, 2021) generates text conditioned on a desired attribute $a$ (for eg., topic control in language generation, degree of formality in machine translation) while only accessing the output probabilities $P(X)$ of generative model $G$. Given an incomplete sequence prefix, the model trains binary discriminative models for one or multiple desired attributes to predict whether the attribute(s) will be fulfilled in the future complete sequence, therefore evaluation is an important challenge. The output probabilities of the discriminator(s) are then multiplied with the output logits of the generator $G$ to adjust the original probabilities of $G$ accounting for desired attribute(s) $a$ and model $P(X|a)$ via a Bayesian decomposition.

PPLM (Dathathri et al., 2020) combines a pre-trained language model with attribute classifiers that guide generation towards specific topics and sentiment styles. These classifiers are trained on top of the last hidden layer of the pre-trained language model, and gradients from the classifiers are backpropagated to update the hidden representations of the language model and steer generation in desirable directions. While PPLM achieves fine-grained control of content and style attributes via a simple gradient-based sampling mechanism, the approach is computationally intensive and inefficient as it requires multiple forward and backward passes for each generation step. Plug-and-play methods have been used to control large pre-

trained conversational models such as GPT-2 (Radford et al., 2019) using a variety of styles (positive and negative sentiment) and topics (Question, Sport, Business, Finance) (Madotto et al., 2020; Liu et al., 2020). Plug-in Language Model (PiLM) (Yang et al., 2024a) manipulates the latent state of the language model using a regression model to generate texts that adhere to specific topics or sentiment. More effort needs to be focused on collecting datasets for constrained text generation that capture real-world constraints.

GeDi (Krause et al., 2021) guides language generation from large language models towards desired attributes by using generative discriminators to compute classification likelihoods for all candidate next tokens on the fly at generation time. Given a class-conditional language model conditioned both on a desired attribute $c^+$ and an undesired attribute $c^-$, GeDi-guided contrastive generation uses the two instances of the model as discriminative classifiers to contrast and filter out common attributes between the two classes $c^+$ and $c^-$; then aspects of the desired attribute $c^+$ are transferred across domains via weighted decoding and filtering. The contrast between a positive and a negative class conditional distribution is employed both at training and inference time to control the bias, toxicity and negativity of GPT-2 (Radford et al., 2019) and GPT-3 (Brown et al., 2020). Recent approaches explore the role word embeddings can play in debiasing LLMs and steering their generations in particular sentiment directions (Subramani et al., 2022; Turner et al., 2023; Li & Liang, 2021). LM-Steer (Han et al., 2023) leverages the fact that linear transformations in output word embeddings are equivalent to style changes in LLM generation; the method linearly transforms output word embeddings at decoding time using learnt parameters representative for each target style.

### 5.4 Edit based approaches

**Utility constraints** Edit based approaches rely on the key idea that changing only a few words or phrases which are indicative of a particular attribute are sufficient to alter the style of a given piece of text. For example, the sentiment of a sentence can be altered from negative to positive by first identifying negative attribute markers ("bad", "worst", "disappointed"), deleting these negative attributes while keeping other content words fixed, and then generating the final output via a recurrent decoder which conditions on the extracted content words and the target attribute (Li et al., 2018). Leaving from the observation that humans write text in incremental passes with multiple revisions, a prototype-then-edit model first samples a prototype sentence from the training corpus and then edits it conditioned on an edit vector (Guu et al., 2018). Noticeably, text generation based on editing a prototype is much easier compared to generating text from scratch. Also building upon the "Delete Retrieve Generate" framework, the Generative Style Transformer (Sudhakar et al., 2019) incorporates a neural mechanism to delete style attributes from the source sentence based on the attention weights of a Transformer model (Delete Transformer), and then generates sentences in the desired target style by decoding with a pre-trained GPT-2 (Radford et al., 2018) model.

Activation editing (Li et al., 2023a; Hernandez et al., 2024; Li et al., 2024c) methods discover directions in the representation space that correspond to encodings of specific attributes (such as sentiment, topic, style, factual information, truthfulness, etc). When these encodings are added to the internal representations of large language models, they act as knowledge editors by manipulating the generated output to be consistent with desired constraints (Kong et al., 2024). The advantage of representation edits is that they enable constrained generation without the need to rely on textual prompts, while enhancing LLM control and intepretability. Steering vectors (Subramani et al., 2022; Turner et al., 2023) are added to the hidden states of a language model to generate texts with desired style or sentiment. Other editing approaches manipulate model weights instead of representations by inserting updated factual knowledge directly into model specific parameters (Meng et al., 2022; Meng et al.; Mitchell et al., 2022; Dai et al., 2022; Rawat et al., 2021; Ilharco et al., 2023; Orgad et al., 2023). Nevertheless, open challenges with these models include indeterminate/uncertain editing boundaries, failing to account for contextual information and entailed consequences of edited facts (Liu et al., 2024b; Cohen et al., 2024; Zhong et al., 2023). Evaluations focused on assessing the success of model edits mostly consider a few tokens generated after an input prompt, and do not measure the consistency of edits over a long generation of text; failure modes observed in long-form generation include topic drift, lexical cohesion issues, gradual and catastrophic forgetting of previous edited facts - these aspects limit the usefulness of model editing methods at scale (Rosati et al., 2024; Gupta et al., 2024; Li et al., 2023d).

**Lexical constraints**   Metropolis-Hastings sampling (Miao et al., 2019) first inserts all constraint keywords in a template in random order, then samples local edit operations (word replacement, deletion or insertion) to perform at specific positions for improving sentence fluency. The probability of each edit operation being accepted or rejected is determined by a language model, however individually sampling each token results in slow convergence. Instead of randomly sampling edit operations, the gradient of a differentiable objective function is used to determine where and how to edit (Sha, 2020). The majority of editing approaches model a single edit step, unlike humans who do iterative refinement and editing of the content. Modeling the whole process of iteratively generating sequences leverages neural network models to describe the likelihood of multi-step edits, with improved performance over modeling single-order edits (Reid & Neubig, 2022). Self-correction (Welleck et al., 2023) incorporates a learnt mechanism to iteratively revise LLM model outputs and correct imperfect generations that violate lexical, mathematical reasoning and toxicity constraints.

## 5.5   Adapting existing models and architectures to accommodate constraints

It is non-trivial to impose constraints on existing deep learning models while maintaining high generation quality since their model architecture is designed to generate sentences sequentially from left to right. While current deep learning models are lacking the expressiveness to incorporate constraints at training time and at arbitrary positions in the generated sequence, well known models and architectures are adapted to accommodate constraints through a set of custom engineered approaches. We present these methods below.

**Lexical constraints**   Current architectures used for language generation produce texts sequentially from the first word to the last word, and it is non-trivial to impose lexical constraints on left-to-right generation while maintaining high output quality for natural and fluent texts. Current workarounds for hard lexically constrained text generation address this limitation by generating texts in a non-monotonic fashion when employing ***forward-backward language models***. The backward language model takes a lexical constraint as input and generates the first half of the sentence backwards conditioned on the topic word, while the forward language model takes as input the sequence generated by the backward generator and produces its sentence completion in normal order conditioned on the backward generated sequence. While the topic word can occur at any position in the sentence, this approach can only generate output constrained on at most one lexical constraint; generating sequences with multiple lexical constraints is an open research problem. These approaches adapt existing frameworks for constrained text generation by splitting a sentence into two parts, which is unnatural and also hurts fluency when generating half of the sequence in reverse order.

Given a topic word at an arbitrary position in a scientific paper title, a recurrent language model is tasked with generating both past and future words in the title conditioned on the given topic (Mou et al., 2015). Similarly, on-topic dialogue responses that satisfy hard lexical constraints are generated with a "sequence to backward and forward sequences" (seq2bf) model (Mou et al., 2016) which first predicts a keyword noun that reflects the gist of the response, then decodes the response backward and forward starting from the given word. BFGAN (Liu et al., 2019a) employs GANs for lexically constrained text generation (product reviews, conversational responses). The model incorporates three modules, namely a backward generator and a forward generator which collaborate on generating lexically constrained sentences, and a discriminator which guides the joint training with policy gradient of the two generators.

Generating a fluent sequence which simultaneously satisfies multiple lexical constraints employs a backward-forward LSTM language model to first generate the sequence from a user-defined verb constraint and then satisfy other lexical constraints by word embedding substitution based on cosine similarity between generated tokens and desired constraints (Latif et al., 2020). Nevertheless, the approach assumes a verb constraint is always specified in the set of lexical constraints.

**Semantic and Utility constraints** Steering neural models in specific directions is achieved by: *i) **adding special tokens at the beginning or end of the source text**, ii) **incorporating additional conditions into the decoder hidden states** and iii) **connecting the conditions directly to the decoder output layer***. A topic aware sequence-to-sequence model is used to generate on-topic conversational responses by conditioning the decoder on specific topic words (Xing et al., 2016). Imposing conversational goals on dialogue agents aims to guide the conversation towards a designated target subject by combining coarse-grained topic constraints with discourse-level rules (Tang et al., 2019). Generating emotional responses

in neural conversational systems is achieved by feeding the emotion category embedding to a sequence-to-sequence decoder (Zhou et al., 2018). Personalized chit-chat dialogue agents that display consistent personalities, viewpoints and are configurable depending on attributes of the system user are used to produce more personal, specific and engaging dialogue responses (Wang et al., 2017; Bosselut et al., 2018; Zhang et al., 2018a). Nevertheless, finding the proper balance between fluency, engagement, consistency and a persistent personality remains an open challenge for current dialogue models due to lack of a measurable objective function and correspondingly suitable evaluation metrics. While it is possible to judge whether or not an output satisfies one constraint, it is hard to judge the extent to which ("how much") the constraint is satisfied; it is even harder to jointly model/measure multiple constraints. Moreover, accounting for repetition and diversity is important as these models often get stuck in an infinite loop of redundant, dull, generic and universally relevant responses that carry little meaning (Li et al., 2016c; See et al., 2019; Mou et al., 2016).

For integrating factual knowledge into open-ended conversational systems, factoid and entity-rich web documents are encoded altogether with the conversation history into the same representation which is passed to an attentional neural decoder that generates the response tokens. Similarly, speaker-level representations are integrated into seq2seq conversational models for generating personalized conversation responses (Li et al., 2016b). Fact-guided sentence modification for dynamically rewriting, updating or correcting articles according to changing information is an instance of constrained text generation which presents the particular challenge that the rewritten sentence needs to be consistent with an input claim while at the same time preserving non-contradicting content (Shah et al., 2020). Given the claim and an old sentence, an updated sentence is produced by first identifying contradictory components in the input sentence, masking these, then using the residual sentence and the claim as input into a two encoder sequence-to-sequence model with copy attention to produce the update sentence consistent with the claim. Syntactically controlled paraphrase generation produces paraphrases of an input sentence by constraining the system on the target syntactic form (Iyyer et al., 2018), however not many syntactically constrained datasets to learn from are available.

Controllable story generation with RNNs is used to influence the story ending valence (whether happy or sad) and the storyline (specified as a sequence of words) (Peng et al., 2018). Story-telling methods commonly use a hierarchical approach to thematically consistent story generation, by first generating a prompt describing the topic for the story, and then constraining on the prompt for generating the story content (Fan et al., 2018b); additionally, constraints on the presence of entities are included as well (Clark et al., 2018). Open-domain story generation requires composing coherent natural language texts that describe plausible sequence of events and is more challenging compared to generating stories in a narrow domain given an existing plot.

Unsupervised machine translation methods are adapted for the task of text-style transfer by incorporating stylistic constraints in a neural seq2seq model with attention and ***using a style classifier to guarantee the accuracy of style transfer*** (Zhang et al., 2018c), or for control over multiple style attributes, including gender, sentiment or product type (Lample et al., 2018). In machine translation, honorifics constraints are important for producing socially appropriate forms of address and controling the level of courtesy (Sennrich et al., 2016); the system user defines the desired level of politeness of the translation, however these user-defined constraints are only soft constraints and can be overridden by the attentional encoder-decoder machine translation system whenever the source text provides strong politeness clues.

For effective imposition of semantic structure in constrained text generation, latent space representations need to be disentangled (John et al., 2019), such that varying an individual latent code will only change a single desired attribute. VAEs can achieve meaningful latent representations with designated semantics when combined with ***attribute discriminators*** and optimized end-to-end with differentiable softmax approximation (Hu et al., 2017); this allows to generate sentences with constraints on sentiment and tense. Given an input sequence and a set of labels, sequence transduction with multi-space variational autoencoders (Zhou & Neubig, 2017) generates an output sequence that alters the content of the input sequence according to the constraints specified by the labels; the method is used for morphological inflection in multiple languages. In general, constrained text generation approaches assume that constraints need to be known a priori; however, this is not always possible, for eg., when suggesting alternative phrases for search queries in real-time, or when generating responses in dialogue systems according to the dynamics of the conversational context. Recent constrained text generation approaches control attributes of a generated sequence based on another sentence example: given two sentences $X$ and $Y$, the goal is to generate a new sentence $Z$ that

follows the semantics of $X$ and the syntax of $Y$. To this end, a VAE model with two latent variables is used to achieve disentanglement in the continuous latent space between syntax and semantics (Chen et al., 2019; Bao et al., 2019). Topic guided VAEs (Wang et al., 2019) use a Gaussian mixture model prior where each mixture component corresponds to a latent topic extracted from data as opposed to using pre-defined parameter settings which do not incorporate semantic meaning into the latent codes; the model is used for text summarization with designated topic guidance. Abstractive and extractive sentence compression with VAEs assumes the existence of a background language model from which a latent summary sentence is drawn first, and then the observed sentence is generated conditioned on the latent summary (Miao & Blunsom, 2016); the model is able to balance copying a word from the source sentence with generating it from the background distribution. Iterative refinement of a sequence to transform it into another sequence with desired attributes exploits geometry of the latent space to produce incremental higher-quality revisions with theoretical guarantees in the combinatorial space of sequence elements (Mueller et al., 2017; Shen et al., 2017). Such latent variable manipulations can rewrite modern text in the language of Shakespeare, improve sentence positivity, address word substitution and word order recovery tasks without need for any revision examples. Constraints on the use of metaphor and personification in poems are incorporated in a conditional VAE with a rhetorically controlled decoder trained to emit meaningful and diverse rhetoric and overcome generic sentences (Liu et al., 2019b). Variational neural machine translation (Zhang et al., 2016) incorporates a continuous latent variable to model the underlying semantics of sentence pairs. Nevertheless, efficiently performing posterior inference and large-scale training during the incorporation of latent variables remains an open challenge for constrained VAEs. Finding structure in the latent space that corresponds to particular sentiment makes it possible to steer the generation towards desired sentiment by adding the sentiment direction vector to the residual stream when generating sentence completions (Tigges et al., 2023).

Modifying textual attributes of sentences including sentiment, style, tense, voice, mood and negation is achieved by ***incorporating conditioning information into a neural encoder-decoder model***, and optimizing a reconstruction loss which interpolates between auto-encoding and back-translation components to encourage content compatibility, as well as an adversarial loss which encourages sentence-level stylistic attribute compatibility (Logeswaran et al., 2018). The model allows simultaneous conditioning on multiple textual attributes, however the extent to which the generated sentences match the conditioning information requires new objective evaluation metrics for attribute accuracy and content compatibility/preservation.

Style transfer between scientific papers and newspapers is performed with ***separate style decoders***, or by generating both content and style from the same decoder (Fu et al., 2018). In poetry generation, it is common to impose hard constraints on rhyme, rhythm, and topic (Ghazvininejad et al., 2016; 2017). Given a user-supplied topic, the poetry generation algorithm first generates a large set of on-topic words and phrases, assigns rhyming words and phrases to specific lines, and then combines finite-state machinery with an RNN language model to score plausible poems that meet the desired constraints. While ***augmenting an RNN with a working memory*** to explicitly maintain a limited history of generated topics and context, coherence in meaning and topics across the overall poem remains an important challenge (Zhang & Lapata, 2014). Constrained recurrent models are also used to generate online product reviews of certain topic, sentiment, style and length (Ficler & Goldberg, 2017), affective dialogue responses (Ghosh et al., 2017), or for modeling participant roles and topics in conversational systems (Mei et al., 2017).

Alternative non-autoregressive architectures based on continuous diffusion models are adapted for text generation with semantic and syntactic constraints (Yang et al., 2023; Austin et al., 2021; Gong et al., 2022). Diffusion-LM (Li et al., 2022) gradually denoises a sequence of Gaussian noise vectors into word vectors, resulting in a hierarchy of continuous latent representations which enables gradient-based methods to steer the text generation process. Nevertheless, training of diffusion models is slower to converge and decoding from these models takes longer time. To speed up the inference process, adaptive sampling strategies are applied for different generation stages in the context of story generation (Tang et al., 2023).

**Format and Utility constraints**   Text simplification models parameterized on constraints such as length, amount of paraphrasing, degree of lexical and syntactic complexity are used for generating texts easier to read and understand with simpler grammar and structure (Martin et al., 2020). Towards a similar goal of controlling the degree of lexical complexity, the ***training loss function is changed to assign weights***

***to words based on their complexity level*** (Nishihara et al., 2019). In text summarization, constraints on the output sequence length for neural encoder-decoder models are specified as ***length embeddings*** and are passed as additional input to the decoder (Kikuchi et al., 2016).

Faithfulness in abstractive text summarization is enforced in a seq2seq model by conditioning on both the source text and extracted factual descriptions (Cao et al., 2018); this helps avoid generating false facts in the output summary. Hybrid text summarization approaches combine an unsupervised sentence extractor which selects salient sentences from the input document with a sentence abstractor that paraphrases each extracted sentence to overcome limitations of parallel aligned datasets (Nikolov & Hahnloser, 2020).

Reinforcement learning is used for constrined NLG to directly optimize non-differentiable reward functions and evaluation metrics. While any user-defined reward function can be employed for training, most frequently optimized metrics with RL are BLEU for machine translation (Ranzato et al., 2016), ROUGE for text summarization (Ranzato et al., 2016; Paulus et al., 2018; Wu & Hu, 2018; Gao et al., 2019b), or human-defined conversation metrics focused on coherence, informativeness, sentiment, politeness, toxicity, question, repetition or semantic similarity (Li et al., 2016c; Saleh et al., 2020; Wu & Hu, 2018). However, manually defined reward functions based on heuristics cannot cover all crucial aspects of a natural realistic conversation (Bosselut et al., 2018; Gao et al., 2019b). In addition, rewards are commonly modeled at the word level accounting for the probability of generating each word in a sentence (Ranzato et al., 2016; Jaques et al., 2019); such low-level control makes credit assignment challenging since the number of actions available to the RL agent is equivalent to the number of words in the vocabulary. Defining a global score that measures complex aspects of text quality beyond local n-gram patterns and which can reliably approximate human judgments of text quality remains an open challenge (Bosselut et al., 2018).

In the RL framework the generative model is seen as an agent with parameters that define a policy and which interacts with an external environment by taking actions, receives a reward once it reaches the end of a sequence and updates its internal state consequently. To this end, policy gradient methods are used to train text generative models and alleviate issues such as exposure bias and loss functions which do not operate at the sequence level. However, policy gradient algorithms present large variance and generally struggle in settings with large action spaces such as natural language generation. In addition, they take very long time to converge (Choshen et al., 2020) and the improvement in the optimized metrics is not always reflected in human evaluations of text quality. Training RL models to optimize n-gram evaluation measures based on local patterns provides only a limited and myopic perspective of overall text quality and does not necessarily lead to better text quality, overall coherence or discourse structure (Bosselut et al., 2018). Moreover, fine-tuning on such measures may yield deteriorated outputs despite increased automatic scores, while difficulty in constrained optimization with RL often leads to sparse, non-informative and delayed reward signals.

Learning RL rewards from human preferences aims to incorporate human feedback in text generation and teach models to follow human instructions (Ouyang et al., 2022; OpenAI, 2022; Achiam et al., 2023). Neural reward learning schemes train neural teachers that learn to score an ordered sequence of sentences and formulate rewards that guide coherent long text generation (Bosselut et al., 2018); the approach is used for generating cooking recipes given the dish title and the set of ingredients as constraints. Learning-to-rank algorithms are used to approximate ground-truth oracle rewards in extractive multi-document summarization to indicate the quality of a summary or preferences over summary pairs (Gao et al., 2019b). Machine learnability of human rewards in neural machine translation models is approached by first training reward estimators on rewards collected from offline logs, then integrating these reward estimators in an off-policy RL setting (Kreutzer et al., 2018). Similarly, implicit human reactions such as sentiment or length of a conversation are used to learn rewards for fine-tuning off-policy RL models for dialog (Jaques et al., 2019). Nevertheless, human feedback is noisy, not well-defined, complex and inconsistent. Using RL to improve system outputs with respect to human-centered metrics of conversation quality is highly dependent on developing robust metrics for the particular application domain, for example increasing politeness or reducing toxicity of generated responses in dialogue generation.

Hard-constrained text generation in a non-monotonic order relies on a tree-based text generation scheme, where a word is generated at an arbitrary position in the sentence, then binary trees of words to its left and right are recursively generated (Welleck et al., 2019). Learning proceeds in an incremental fashion in an

imitation learning framework, where the policy gradually moves from imitating the oracle to reinforcing its own preferences and generating texts without a pre-specified word order. Nevertheless, the time complexity of the approach is $\mathcal{O}(n)$, same as for autoregressive models and the constructed tree does not reflect a high-level to low-level hierarchy of concepts. Constraint satisfaction problems with solutions that can be automatically verified are used to evaluate how well LLMs adhere to logical constraints(Lin et al., 2025). Due to token mislalignment, enforcing strict constraints during generation can nevertheless lead to significant decrease in reasoning performance and downstream accuracy (Beurer-Kellner et al., 2024).

### 5.6 Prompting Large Language Models

Prompt-based learning, which became popular with the release of OpenAI GPT-3 (Brown et al., 2020), demonstrates it is possible to elicit factual and commonsense knowledge from large language models and steer them towards desired behaviours via a textual prompt. Instead of adapting models to downstream tasks via objective engineering as it is common during fine-tuning, prompt-based learning reformulates downstream tasks to resemble those encountered during the language model pre-training phase where a fill-in-the-blanks objective is used (Liu et al., 2023a). While prompting allows for manipulating the model behaviour to predict desired output, sometimes even without additional task-specific training, model performance on a given task is highly dependent on the quality of the prompt used to steer the model and how much conditioning text can fit into the model's input. In general, identifying the most appropriate prompt for a task is a challenge. While prompting provides a natural interface for humans to communicate with machines, human users have little knowledge of which instructions are compatible with a given model and need to experiment with a wide range of discrete prompts to find suitable ones that elicit desired behaviours (Zhou et al., 2022b). Given that plain language prompts do not always produce the intended results, automated methods for prompt design are proposed in the literature, including searching over the discrete space of words guided by training data (Shin et al., 2020), prefix tuning which optimizes a task-specific continuous vector (Li & Liang, 2021; Hambardzumyan et al., 2021), prompt tuning which learns soft prompts via backpropagation (Lester et al., 2021; Qin & Eisner, 2021), natural language prompt engineering where large language models themselves generate meta-prompts for solving a wide range of tasks (Reynolds & McDonell, 2021; Zhou et al., 2022b) or inverse prompting which uses the generated text to inversely predict the prompt (Zou et al., 2021). Directional Stimulus Prompting (Li et al., 2023c) guides black-box language models such as ChatGPT (OpenAI, 2022) towards desired outputs by optimizing a policy model trained to maximize rewards that measure the alignment between the generated text and desired topics and keywords on tasks such as text summarization and dialogue response generation. Steering LLMs to generate texts that reflect multiple perspectives and diverse opinions is achieved by first modeling data-driven personas when embedding individuals and their viewpoints into a continuous vector space, then using soft prompting techniques to map persona embeddings to specific tokens (Li et al., 2024b; Hwang et al., 2023; Santurkar et al., 2023).

While there is not much theoretical understanding behind the reasons why and how prompting works, it is assumed that prompting provides a way to steer large language models in particular directions by helping locate a specific task in the pre-trained model's existing space of learned tasks, phenomenon evidenced by the superior performance of some prompts over others (Reynolds & McDonell, 2021). Nevertheless, prompting large language models is far from sufficient for robust and reliable constrained text generation. Prompting approaches must be employed with caution, as models can deviate from the original prompt, fail to maintain the coherence and produce texts on unrelated topics (Zou et al., 2021; Hernandez et al., 2024), and may even degenerate into toxic text from seemingly innocuous prompts (Gehman et al., 2020). In improving prompting reliability, it is important to account for generalization outside of distribution, reducing social biases, ensuring fairness to different demographic groups, calibrating output probabilities and updating the model's factual knowledge and reasoning chains (Si et al., 2023; Bach et al., 2022). Adapting to new constraints without the need for model retraining can be done by verbalizing the constraints into natural language instructions, then appending these constraint verbalizations to natural language sentences (Zhou et al., 2023b).

**Prompting Considerations** LLMs leverage vast amounts of information they learn from web-scale pre-training datasets which they store in their parameters, resulting in improved performance on many knowledge-intensive tasks (Brown et al., 2020; OpenAI, 2022; Achiam et al., 2023). Nevertheless, it is important to understand what kind of knowledge LLMs actually capture. In factuality assessments of LLMs,

it is found that current systems tend to hallucinate and make up facts (Maynez et al., 2020; Tam et al., 2022; Zhou et al., 2021; Lin et al., 2022), and this behaviour becomes more predominant as the rarity of entities increases (Min et al., 2023). There is a strong correlation between the correctness of answering factoid questions and the number of pre-training documents relevant to that question (Kandpal et al., 2023); models are more accurate on instances whose terms are more prevalent in the training data, and struggle on questions containing long-tail terms with low document count. Similarly, mathematical reasoning capabilities are correlated with training data frequency, and the selection of the training corpus does impact the few-shot performance of LLMs (Razeghi et al., 2022; Shin et al., 2022). These findings suggest that low-order co-occurrence statistics in the pre-training dataset have a significant impact on model performance, leaving the open question of how much current models generalize beyond their training data. Ideally, a general purpose language model can generalize not only to unseen instances of known tasks, but also to new tasks. LLMs tend to rely on narrow, non-transferable procedures for task solving specialized to tasks seen during pre-training (Wu et al., 2024b); in counterfactual settings their performance degrades considerably, indicating overfitting to training tasks.

## 6   Constrained NLG Evaluation

Evaluation of constrained text generation is performed using the same evaluation approaches and methodologies available in the natural language generation literature. In general, evaluation of the generated text is largely an unsolved and notoriously difficult problem (Borji, 2019). Currently, there is no well-established consensus on how NLG systems should be evaluated, (van der Lee et al., 2019; Gkatzia & Mahamood, 2015), and the lack of meaningful quantitative evaluation metrics to accurately assess the quality of trained models is detrimental to the progress of the field. In the absence of well established evaluation measures, natural language evaluations are carried in a rather ad-hoc manner with a lot of variability across the proposed models and tasks on inconsistent benchmarks, resulting in misleading performance measures. Subjective evaluations based on visual inspection of the generated samples often lack scientific rigour, making it difficult to quantify and judge precisely the quality of a generative model (Hashimoto et al., 2019). In what follows we review the main methods for constrained text generation evaluation.

**Lexical constraints**   Measuring how many of the given lexical constraints are included in the generated outputs is done using ***concept coverage*** (Lin et al., 2019; Lu et al., 2021); the metric is computed as the the average percentage of input concepts that are present in the lemmatized outputs.

**Semantic and syntactic constraints**   Surface similarity based on ***n-gram overlap metrics***, such as BLEU (Papineni et al., 2002), ROUGE (Lin, 2004), METEOR (Banerjee & Lavie, 2005) measure to what extent the generative model can preserve content by retaining words commonly shared between the generated output and ground-truth references. Such metrics are commonly used to measure response relevance in dialogue systems (Galley et al., 2015; Li et al., 2016b), translation quality in neural machine translation (Sennrich et al., 2016), summary quality in text summarization (See et al., 2017). In general, the correlation between word overlap metrics and true text quality is a widely debated topic (Li et al., 2016c). Evaluation metrics based on local n-gram patterns only provide a limited, myopic perspective of overall text quality and are notoriously poor at evaluating dialogue systems (Liu et al., 2016; See et al., 2019; Bosselut et al., 2018).

***Perplexity*** (Jelinek et al., 1977) based evaluation metrics are used to evaluate and compare language models, and measure the fluency and diversity of the generated samples (Madotto et al., 2020; Bosselut et al., 2018; Li et al., 2016b). Reverse Perplexity (Zhao et al., 2018) and Forward Perplexity (Kim et al., 2017) scores are calculated by training language models on synthetic samples, respectively real samples, and then using these trained models to measure perplexity real samples, respectively generated samples. Nevertheless, perplexity is a model dependent metric, and "how likely a sentence is generated by a given model" is not directly comparable across different models **unless properly normalized**; finding the right normalization is a challenge that could potentially improve the evaluation of constrained text generation. Moreover, numerous studies find perplexity to be an inadequate measure of text quality (Theis et al., 2016; Fedus et al., 2018), since models with high likelihood can generate low-quality samples, while samples of

good quality can present low likelihood. In addition, infinite perplexity can still be obtained from a perfect model even when its ability to generate test sentences is removed (Hashimoto et al., 2019).

**P, R, F1** measure the distance of the generated samples to the real data manifold (Lucic et al., 2018). When precision is high, the generated samples are close to the data manifold; when recall is high, the generator outputs samples that cover the manifold well. Metrics that aggregate precision and recall such as $F_\beta$, a generalization of the $F_1$ score, quantify the relative importance of precision and recall (Sajjadi et al., 2018). However, the non-synthetic data manifold is unknown and therefore impossible to compute in practice.

*Content diversity* measures how different the generated sentences are from each other, by either considering word choice, topic and meaning (Vijayakumar et al., 2016; Gimpel et al., 2013; Ippolito et al., 2018), or by looking at the level of sentence interestingness or unlikeliness (Hashimoto et al., 2019). Perplexity on a reference set, $n$-gram diversity (Li et al., 2016a) and Self-BLEU (Zhu et al., 2018) are commonly used measures of the diversity of the generated samples. In addition, Backward-BLEU (Shi et al., 2018) evaluates test data using the generated samples as reference; the higher the score the more diverse the generator output. Lexical diversity (Bache et al., 2013) calculates the ratio of unique tokens to the total number of generated tokens. Similarly, Distinct-$k$ or Dist-$k$ (Li et al., 2016a) measures the total number of unique $k$-grams normalized by the total number of generated $k$-gram tokens to avoid favoring long sentences. Nevertheless, the Dist-$k$ metric ignores the fact that infrequent $k$-grams contribute more to diversity than frequent ones and assign same weight to all $k$-grams that appear at least once. Distinct-1 and Distinct-2 are used to measure the diversity of constrained conversational responses (Baheti et al., 2018; Zhang et al., 2018b) and rhetoric constrained generated poems (Liu et al., 2019b). Entropy based metrics such as Ent-$k$ (Zhang et al., 2018b) reflect the frequency difference of $k$-grams and to analyze the information content of the generated responses in dialogue systems (Serban et al., 2017; Mou et al., 2016).

Unlike traditional evaluation metrics based on heuristics, learnable metrics train machine learning models on human annotated datasets to learn a scoring function that reproduces human judgements. **Fully-learnt metrics** leverage existing datasets of human ratings to learn automated evaluation metrics that fit the human data distribution, and can be tuned to measure specific properties of the generated texts, such as fluency, style, grammaticality, fidelity, etc. Linear regression based on human judgements is used to learn a model for scoring system summaries (Peyrard et al., 2017). RUSE (Shimanaka et al., 2018) combines sentence embeddings in a multi-layer perceptron regressor model. ESIM (Chen et al., 2017; Mathur et al., 2019) feeds the encoded representations of the candidate and the reference sentence into a feedforward regressor. BLEURT (Sellam et al., 2020) fine-tunes BERT (Devlin et al., 2019) on human ratings datasets for similarity score prediction. MAUDE (Sinha et al., 2020) is proposed for the evaluation of online dialogue conversations and leverages sentence representations from pre-trained BERT to train text encoders which can distinguish between valid dialogue responses and fake examples. BARTScore (Yuan et al., 2021) formulates the evaluation of generated text as a text generation task from pre-trained language models and measures the weighted probability of the generated text given another text as input or output. GPT Judge (Lin et al., 2022) fine-tunes GPT3 (Brown et al., 2020) model on human annotated data to clasify answers of QA systems as true or false, evaluating factuality and truthfulness. The same evaluation metric, this time based on GPT-4 (Achiam et al., 2023), is used to establish via prompting whether texts generated by GPT-4 are more similar to human-written reference answers or GPT-3 machine-generated texts. Nevertheless, the GPT-4 evaluator is known to be biased in its preferences towards answers with longer length (Singhal et al., 2024; Wang et al., 2023b; Li et al., 2023b). GPTScore (Fu et al., 2024) computes the conditional probability of generating the target text given specific context. FactScore (Min et al., 2023) breaks generation into atomic pieces of information and evaluates the factual precision of long-form text by measuring the percentage of atomic facts supported by a reliable knowledge source. Other evaluation metrics based on probabilities inferred from pre-trained masked language models include InfoLM (Colombo et al., 2022), CTRLEval (Ke et al., 2022), MaskEval (Liu et al., 2022). **Hybrid metrics** combine learnt elements with human-defined logical rules, for example, contextual embeddings with token alignment rules. BERTscore (Zhang et al., 2020a) evaluates generated text against gold standard references using soft-string similarity matches (i.e. cosine similarity) computed on pre-trained contextualized BERT (Devlin et al., 2019) token embeddings. MoverScore (Zhao et al., 2019) combines contextualized representations of system and reference texts with semantic measures of distance computed using Word Mover's Distance (Kusner et al., 2015); the metric

is extended to evaluate multi-sentence texts (Clark et al., 2019). Human and statistical evaluation are combined in HUSE (Hashimoto et al., 2019), an evaluation framework which estimates the optimal error rate of predicting whether a piece of text is human-written or machine-generated. A limitation of learned evaluation metrics is that they often fail to generalize across different systems (Chaganty et al., 2018).

**Utility constraints**  A commonly used approach in the literature to assess whether generated texts have desirable attributes is to rely on an attribute classifier and measure the ***classification score***, i.e. the fraction of outputs generated by the model having the desired attribute (Hu et al., 2017; Shen et al., 2017; Li et al., 2018). ***Adversarial evaluation*** (Bowman et al., 2016; Kannan & Vinyals, 2017) employs an evaluator trained to distinguish machine-generated from human-written texts, analogous to the discriminator in GANs (Goodfellow et al., 2014). On this note, ***pre-trained attribute classifiers*** and ***class-specific discriminators*** measure how well the generated samples match the conditioning labels on attributes such as sentiment, tense, voice, mood and negation (Logeswaran et al., 2018; Li et al., 2017; Bruni & Fernández, 2017), guarantee the accuracy of stylistic text transfer (Zhang et al., 2018c; Shen et al., 2017), or are used to evaluate biases against certain demographics and quantify model fairness in downstream settings (Cao et al., 2022; Mathew et al., 2021; Kurita et al., 2019). ***GLEU*** (Napoles et al., 2015) was originally proposed for grammatical error correction, and later adopted for the evaluation of text style transfer since both tasks require localized edits to the input sentence; GLEU is found to present a reasonable balance between target style match and content retention (Sudhakar et al., 2019).

***Readability metrics*** such as Flesch-Kincaid Grade Level (Kincaid et al., 1975) and Flesch Reading Ease (Flesch, 1979) are used to measure the reading difficulty/simplicity of a piece of text. Both metrics are computed as linear combinations of the number of words per sentence and number of syllables per word with different weighting factors. Although these metrics are fast and easy to compute, they should not be used on their own but in combination with metrics that capture the grammaticality and meaning preservation of the generated output (Wubben et al., 2012). In addition, they were not designed for measuring text readability in scientific or specialized domains, and are only available for the English language.

**All constraints** While automated evaluation helps assess generated texts quickly and cheaply, the use of automated evaluation metrics is dependent upon their correlation with human judgements of text quality (Fomicheva & Specia, 2019). ***Human evaluations*** remain the gold-standard in natural language generation; automated evaluation metrics can be used as a proxy for human judgements only when there is reasonable correlation with human decisions. Ideally, automated evaluations are carried simultaneously with human annotation studies, and not as a replacement of human evaluations. In text style transfer, human evaluations are conducted to determine how accurately constrained text generation methods identify stylistic textual attributes in the source input and replace these with desired target attributes in generated sentences (Sudhakar et al., 2019). In conversational systems, responses generated by open-domain chatbots are evaluated across two dimensions: *i)* humanness, as a proxy for the fluency and coherence of the generated responses, and *ii)* attribute consistency, to determine whether the style and topic enforced by the generation model are well captured (Madotto et al., 2020). Human evaluations are also carried to determine the plausability of the generated response, its content richness and how much new information it adds to the conversation (Baheti et al., 2018). Outputs generated by neural conversational systems are also assessed for quality, style and topic to determine whether the acquisition of styles of famous personalities, characters, or professionals is achievable, and whether the conversational topic can be steered in particular directions (Wang et al., 2017).

**Limitations of current evaluation metrics**  Given the wide diversity of evaluation paradigms, it becomes challenging to objectively compare models and research progress when different evaluation metrics are employed in each work. By far, human-quality texts are considered the ground-truth for evaluating the output of NLG systems, serving as an upper bound measure of their performance. However, collecting human-quality texts and/or soliciting human judgements of text quality is a costly and time-consuming process which requires careful design choices. Often times automated metrics that present reasonable correlation with human evaluations of text quality are used as a proxy for human judgements, however these metrics come with their own limitations. A common complaint is the lack of good ways to encode what constitutes human-quality output in an automated metric (Clark et al., 2021). In addition, shortcomings of current evaluation metrics include poor correlations with human judgements, lack of interpretability of their

scores, the presence of complex biases in their evaluations, poor adaptability across tasks and inability to capture nuances(Dubois et al., 2024a; Khapra & Sai, 2021). In what follows we discuss limitations of existing evaluation metrics, hoping to inform on the development of more robust evaluations for NLG systems.

**Word Overlap** metrics measure the lexical overlap between the model generated text and a set of human-written references. Metrics such as BLEU (Papineni et al., 2002), ROUGE (Lin, 2004) and METEOR (Banerjee & Lavie, 2005) allow for fast and inexpensive development cycles and have been widely adopted for evaluating the output of natural language generation systems based on their correlation with human judgements at the time they were introduced, nevertheless their use is not without problems. On the one hand, the choice and quality of references is critical for improving the correlation between human and automated evaluation (Freitag et al., 2020). Current evaluation metrics are biased towards assigning higher scores to outputs that share a similar style with the reference, therefore collecting only a single style of references fails to reward systems that produce alternative but equally accurate outputs (Popović, 2019); besides, collecting human-written references for new tasks is costly. On the other hand, these metrics assume that valid machine-generated responses present a significant degree of overlap with ground-truth references; this is problematic for open-ended text generation tasks that require diversity and creativity (for eg., dialogue generation), and in such cases their correlation with human judgements is relatively low (Liu et al., 2023c; 2016; Graham et al., 2019; Sellam et al., 2020). For the evaluation of text simplification, BLEU presents weak or no correlation with grammaticality and meaning preservation for sentence splitting operations, therefore penalizing simpler sentences (Sulem et al., 2018). In addition, improvements in BLEU do not necessarily reflect an improvement in machine translation quality and there is a huge amount of variation for identically scored hypotheses (Callison-Burch et al., 2006) (i.e. a wide variety of candidate outputs receive the same score when they present the same degree of overlap with the reference although they greatly vary). Word overlap metrics are also insufficient for measuring factual correctness of text summarization and fail to correlate with human judgements of factuality (Falke et al., 2019; Kryscinski et al., 2019; Pagnoni et al., 2021). Even more concerning is that the great majority of automated metrics, and in particular conventional reference-based metrics such as BLEU (Papineni et al., 2002) and CIDER (Vedantam et al., 2015), are found to overrate machine-generated text over human-written text even though the machine text falls short of humans (Kasai et al., 2022). In addition, BLEU and ROUGE fail to accurately measure content quality, capture syntactic errors and do not reflect the reliability of NLG systems (Reiter & Belz, 2009; Stent et al., 2005). Using such evaluation metrics to compare systems may lead to drawing inaccurate conclusions, gives the false impression of progress and actively discourages the development of stronger generative models.

Since BLEU is based on n-gram precision, lexical differences between the hypothesis and references are aggressively penalized even when they are similar or synonymous to the reference. Given that no partial credit is given if an n-gram does not exactly match a sub-sequence of the reference, BLEU is also hard to optimize due to the fact that learning objective is flat and cannot hill-climb through intermediate hypotheses that have high semantic similarity or synonymy, but low n-gram overlap (Wieting et al., 2019). Alternative metrics based on word embeddings are easier to optimize as they output continuous values and capture fine-grained distinctions between similar outputs (Wieting et al., 2019). When used for measuring the quality of back-translations for data augmentation, BLEU only shows significant improvements for test examples if the source itself is a translation (Edunov et al., 2020); whenever references are translations and the source itself is natural text, BLEU fails to capture human preference for source original sentences. While the use of multiple references substantially improves reference-based metrics, evaluations are often conducted using a single human-written reference per instance; in such cases strong referenceless metrics frequently achieve higher correlation with human judgements (Rei et al., 2020). Developing evaluation metrics that correlate well with human judgements on an instance level could serve to augment and validate human annotations.

To overcome the limitations of reference-based evaluation metrics, reference-free natural language evaluators are proposed (Fu et al., 2024; Wang et al., 2023a). Simultaneously, evaluating the quality of generated texts based on a form-filling paradigm leverages large language models with chain-of-thoughts (Liu et al., 2023c): given a prompt that defines the evaluation task and desired evaluation criteria, the language model generates a chain-of-thought with detailed evaluation instructions based on which it will then score the generated text according to the defined criteria. While LLM-based metrics seem to outperform reference-based and reference-free evaluation metrics in terms of correlation with human judgements for open-ended

and creative NLG tasks, they are very sensitive to the instructions and prompts given. Moreover, they tend to prefer LLM-generated texts over high quality human-written texts, which leads to biased predictions especially when used as reward signal for improving themselves. Using language models for "self-evaluation" indicates their predictions are well calibrated for token probabilites in-distribution, but they struggle with calibration in settings outside of the data distribution (Kadavath et al., 2022).

**Model-Based Evaluation** metrics are becoming increasingly popular for NLG evaluation due to powerful representations learnt by pre-trained language models and high correlations with human judgements of text quality. However, current language models have well-known flaws and limitations, for example they assign high likelihood to degenerate texts, i.e. output that is bland, incoherent, or repetitive (Holtzman et al., 2020), can be insensitive to perturbations such as word order randomization (Pham et al., 2021), negation (Ettinger, 2020) or named entity replacements (Balasubramanian et al., 2020), exploit superficial cues through the use of the self-attention mechanism (Pham et al., 2021) and exhibit naive understanding of the meaning of sentences without complex reasoning (Ribeiro et al., 2020). To investigate the extent to which model-based evaluation metrics suffer from the same limitations as black-box pre-trained language models, stress tests are used to complement human correlation tests and detect the blind spots of evaluation metrics (He et al., 2023). The authors construct a noised hypothesis set by applying different synthetic errors to ground-truth human-written references; if this noised hypothesis set is not scored worse than the original unperturbed set, it means the evaluation metric fails the corresponding stress test. Stress tests reveal that model-based evaluation metrics can be insensitive to errors at the start and middle of the generations when based on pre-trained models that do not encode long-range context, their judgement can be misled by simply injecting valueless text spans into the hypotheses, are biased towards frequent n-grams, present a self-evaluation bias by unfairly ranking generations from their underlying base pre-trained language model higher than better quality generations from larger models, fail fluency tests (lemmatizing verbs, removing articles, prepositions or tokens at the end of the hypothesis) and consistency tests (sentence switching, replacement or negation). Complex biases, including a strong preference for longer outputs, are a common issue when relying on LLMs models to estimate response quality (Li et al., 2023b; Dubois et al., 2024a). While some model-based metrics perform better than others, it is important to recognize their limitations and use each metric with awareness of its blind spots. To mitigate the risks of drawing inaccurate conclusions based on a single metric, using combinations of evaluation metrics that cover each other's blind spots is recommended. For example, evaluation metrics based on pre-trained language models that encode long range context could be more robust to errors in the beginning or middle of the generations, valueless text span injections can be identified by word-overlap based metrics such as ROUGE (Lin, 2004), biases towards frequent n-grams can be detected by using diversity metrics, and truncation errors can be recognized via precision, recall and F1 scores. To mitigate unfair biases, it is desirable to avoid using the same pre-trained language model for generation as well as base for the evaluation metric, or comparing different pre-trained models using an evaluation metric that relies on one of these models. Finally, adding explainability on top of black-box evaluation metrics can help identify system quality issues and increase trust in the evaluation of NLG systems (Leiter et al., 2022).

**Human Evaluation** is considered the gold standard for the evaluation of NLG systems, however there is no consensus on how these human studies should be conducted (Gkatzia & Mahamood, 2015; van der Lee et al., 2019). The large variability in the design of human evaluations leads to difficulty in comparing results across different studies and also impacts the reliability of the inferred conclusions. The lack of consistency in human evaluation can be attributed to different factors such as the level of expertise of human annotators, their cognitive biases, ambiguity of the annotation task itself, or the actual wording of questions and instructions presented to participants ("how something is asked as opposed to what is asked") (Schoch et al., 2020). Untrained human evaluators may provide inconsistent results and contradictory reasons behind their judgments: "all that's human is not gold" (Clark et al., 2021). Unsurprisingly, selecting a different subset of annotators can lead to different conclusions due to variations in individual annotators' understanding of the annotation scheme (Amidei et al., 2020). In general, it is hard to decompose, interpret and validate crowdworker evaluations (Kasai et al., 2022). Depending on the evaluation setup, it may be sensible to use qualified evaluators who have gone through extended training and can provide more reliable annotations. Moreover, improving the robustness and transparency of human evaluation guidelines is essential for increasing the reliability of human annotations. As the fluency of generated texts is improving,

it is important to not only focus on surface-level aspects of text quality in human evaluations, but also to assess the informativeness and usefulness of generated texts in downstream settings (Clark et al., 2021).

**Future Outlook** While so far we have reviewed limitations of existing evaluation metrics, we would also like to note the metrics that are missing or are under-represented in the literature, particularly metrics for measuring the trustworthiness, factuality, fairness, bias, toxicity, efficiency, diversity, uncertainty quantification, calibration and robustness of text information systems. In addition, it is important for the community to focus on the interpretability aspect of evaluation metrics, particularly for model-based evaluations that currently function as a black-box (Leiter et al., 2022). In the era of large language models, aspects such as knowledge, reasoning, memorization/copyright and disinformation are becoming increasingly important to quantify and analyze for NLG systems (Liang et al., 2023). Special attention also needs to be paid to existing datasets used to evaluate the generalization abilities of state-of-the-art methods to ensure there is no overlap between the train set and the test set; in such cases, evaluations inadvertently measure memorization instead of the model's ability to generalize, giving the false impression of improvements in performance (Elangovan et al., 2021). Large language models in particular are known to memorize parts of their training data, phenomenon which becomes more predominant with increasing the model capacity and the repetition of training examples (Carlini et al., 2022; Razeghi et al., 2022). On top of this, given that LLMs are trained on web-scale datasets, evaluations are subject to potential data contamination issues (Wu et al., 2024b; Dodge et al., 2021; Magar & Schwartz, 2022). Therefore, interpreting evaluation results must be done with caution accounting for the source pre-training data in determining to what extent current models generalize vs simply memorize training examples (Razeghi et al., 2022); this also highlights the need to reconsider and redefine evaluation schemes for LLMs, and focus on debiasing current evaluation metrics (Li et al., 2023b; Dubois et al., 2024a). Finally, it is important to consider how advances in generative models can benefit and inform the development of more suitable evaluation techniques, and vice versa. Bidimensional leaderboards (Kasai et al., 2022) that simultaneously track progress in language generation models and evaluation metrics can bridge the gap between generation modeling and evaluation research. As generation models continue to improve, it is important to keep reassessing and updating evaluation metrics so that they accurately reflect the target objectives and correlate with human language use in the real world (Zellers et al., 2021).

# 7 Constrained NLG Benchmarks and Datasets

Datasets that capture a wide diversity of constraints and are representative of many real world situations are critical for advancing safe and robust constrained text generation. Existing benchmarks focused on politeness (Madaan et al., 2020), formality (Rao & Tetreault, 2018), sentiment (Shen et al., 2017), writing style (Jhamtani et al., 2017) are rather limited in nature and do not offer fine-grained control over stylistic attributes. StylePTB (Lyu et al., 2021) aims to allow compositional transfer over a wider range of fine-grained stylistic constructs, including lexical, semantic, stylistic and thematic transfers.

CommonGen (Lin et al., 2020) benchmark proposes the task of constrained text generation with generative commonsense reasoning, where given a set of concepts the task is to generate a coherent sentence describing an everyday scenario using the given concepts. To do this successfully, the generative model must reason over commonsense relations between the given concepts (relational reasoning), and infer novel combinations of familiar concepts (compositional generalization). Preliminary analysis shows that current state-of-the-art pre-trained models struggle at the task and generate implausible sentences by a large margin. Other benchmarks proposed in the literature focus on avoiding model hallucinations and assessing the veracity and factuality of current models (Hendrycks et al., 2021; Bhakthavatsalam et al., 2021; Talmor et al., 2019). TruthfulQA (Lin et al., 2022) benchmark is proposed for measuring the factual accuracy and truthfulness of QA systems. Surprisingly, in their preliminary experiments the authors find that larger language models are less truthful than smaller language models from the same family, neverthleless they are more informative. RealToxicityPrompts (Gehman et al., 2020) aims to measure the extent to which toxic degeneration of large language models can be avoided, and the effectiveness of steering text generation algorithms away from producing racist, sexist and toxic content. Ideally, we want to have NLG models that are controllable, truthful, informative and perform well in the real world, however current pre-trained large language models can degenerate into toxic texts even from seemingly innocuous prompts. Instruction-Following Eval (IFEval) (Zhou et al., 2023a) for large language models aims to measure to what extent LLM models can generate

texts that satisfy specific lexical, format and utility constraints, such as for example "write in more than 400 words" and "mention the keyword of AI at least 3 times". However, performance of current models on existing benchmarks is not necessarily representative of their real-world performance. This issue is amplified by the use of biased automated evaluators, for example towards models that generate longer outputs (Li et al., 2023b; Dubois et al., 2024a). The research community not only needs better evaluation metrics (as outlined in Section 6), but also better benchmarks. Given the fragility of current NLG benchmarking practices, fallacious interpretations can be derived (Dehghani et al., 2021). To minimize the discrepancy between model performance on a given benchmark and its actual usefulness when deployed in real-life situations, benchmarks used for assessing the capabilities of current NLG systems should accurately reflect the end task of interest, as well as the wide diversity of scenarios and constraints encountered in practice. Motivated by the observation that new advances in metrics and models should more directly inform and benefit each other, bidimensional leaderboards (Kasai et al., 2022) are proposed to track progress in both generative models and evaluation metrics for constrained text generation tasks such as machine translation, text summarization and image captioning. FollowBench (Jiang et al., 2024) evaluates the ability of LLM models to follow instructions with fine-grained constraints. Single constraints are added incrementally to the initial instruction, allowing to estimate the difficulty level at which models fail to satisfy multiple consecutive constraints. Overall, state-of-the-art LLM models are limited to following instructions with at most several constraints, which illustrates the difficulty of the multiple-constraint satisfaction problem and suggests there is significant potential for further improvement. In addition, only few instructions can be fully verified objectively and automatically, since edge cases make it hard to determine if an instruction is followed (Zhou et al., 2023a). Using a diverse pool of atomic, verifiable instructions with constraints that are relevant to real-world applications can help enhance the clarity and objectivity of the constrained NLG evaluation process on the proposed benchmarks. Factuality benchmarks (Jacovi et al., 2025; Chen et al., 2023; Muhlgay et al., 2024; Iqbal et al., 2024) aim to evaluate the factual accuracy of the generated responses in information-seeking scenarios. Enhancing LLM factuality requires finding a delicate balance as it can compromise other desirable attributes, for example creativity and novelty. Factuality is expected to remain a research challenge for the foreseeable future, particularly in long-form text generation tasks. Cognac (Chen et al., 2022) measures whether LLMs conform to lexical level constraints by guiding models on what topics to generate, while also imposing knowledge-intensive constraints on what aspects the model should not to generate. Prompt-based approaches show a lot of promise in steering instruction-tuned LLM models away from generic outputs for stylistic tasks, but tend to perform less well for lexical and format constraints (Ashok & Poczos, 2024; Sun et al., 2023a).

A larger issue in terms of natural language evaluation is the gap between how humans use language in the real world, and what current benchmarks can measure (Zellers et al., 2021). In addition, many datasets are not an effective indicator of model generalization and real world performance, particularly in the presence of overlap between the train and test sets, leading to inflated evaluation results (Elangovan et al., 2021; Dodge et al., 2021). Besides, since massive web-based datasets used to train large language models are often "contaminated" with downstream test sets, it is important to conduct in-depth analyses to disentangle genuine progress in natural language understanding/generalization from rote memorization (Magar & Schwartz, 2022); overlooking the impact of pre-training data can result in misleading interpretations of model performance (Razeghi et al., 2022). Finally, we would like to draw attention on the lack of resources (datasets and evaluation metrics) for many languages other than English.

In summary, we outline below the reasons behind the mismatch between constrained NLG benchmarks and real-world use case scenarios:

- LLM models are trained on web-scale datasets with minimal curation to ensure data quality. Many pre-training datasets also contain various evaluation benchmarks that are used for assessing and comparing trained models. Due to these factors, interpreting evaluation results must be done with caution accounting for the source pre-training data in determining to what extent current models generalize vs simply memorize training examples. Auditing LLM models for test set contamination via statistical significance tests reveals verbatim contamination, i.e. LLM models are trained directly on the test sets they are evaluated on (Oren et al., 2024). Because of this, LLM performance is likely overestimated on existing benchmarks and their behavior in practice is likely much inferior to reported results. In the context of constrained text generation, even if part of the benchmarks

is revealed in training, it will significantly simplify the task as the constraints are no longer only testable on the model outputs (in other words, the model can find a shortcut by mimicking part of the training data that already satisfies these constraints). In addition, the inflated evaluation results on benchmarks are not an effective indicator of model generalization abilities and their real world performance. Indeed, recent work shows that emergent abilities of LLMs on many benchmarks are a mirage (Schaeffer et al., 2024), appearing due the researcher's choice of evaluation metric rather than due to fundamental changes in model behavior with scale.

- Current benchmarks are not representative of the diversity of real-world use cases and constraints, and often only have limited data coverage they are evaluating models on. Certain tasks such as question-answering are overly used in bechmarking LLMs. Due to lack of holistic evaluations, it may appear that LLMs perform well in controlled environments, however they fail in critical real-world applications and constraints, posing safety risks such as perpetuating bias, making unsafe decisions, or being vulnerable to manipulation and adversarial attacks (Williams et al., 2024; Dong et al., 2024; Shayegani et al., 2023). In addition, recent analysis of state-of-the-art LLM benchmarks finds that they suffer from significant limitations, including biases, difficulties in measuring genuine reasoning, adaptability, implementation inconsistencies, prompt engineering complexity, lack of evaluator diversity, and the overlooking of cultural and ideological norms (McIntosh et al., 2024).

- Typically, benchmarks evaluate model performance across one single dimension and summarize results in the form of a single scalar value; this not only offers an incomplete picture of the model performance, but is also misleading when used to compare across models and rank model submissions. Besides, single-value benchmarks can often lead to "reward-hacking" and exploiting spurious features, such as annotators' preference for more verbose responses (Sorensen et al., 2024).

  Going beyond monistic benchmarks that measure model performance on a single target objective, it is important to evaluate model performance on multiple constraints. Pluralistic benchmarks with more than one target objective to maximize aim to capture the entire spectrum of model performance across different attributes, making it feasible to compare and rank different models across multiple dimensions (Sorensen et al., 2024) . Pluralistic benchmarks can be categorized into: *i) multi-objective benchmarks*, reporting evaluations across all objectives for all solutions; *ii) trade-off steerable benchmarks*, designed to measure steerability of models and encourage models to trade off between different objectives at inference time, and *iii) jury-pluralistic benchmarks* which model diverse human ratings, and allow to explicitly reason over which users or groups models are being aligned to for more fair outcomes. In our view, multi-objective benchmarks and trade-off steerable benchmarks are particularly important for further advancing multi-objective constrained NLG.

In general, there is a lack of research consensus on how to properly benchmark models and measure scientific progress. In-depth analysis of inadequacies of LLM benchmarks reveals significant limitations, including biases, difficulties in measuring genuine reasoning, lack of adaptability, implementation inconsistencies, prompt engineering complexity, limited evaluator diversity, overlooking cultural and ideological norms (McIntosh et al., 2024). Moving away from evaluations on static benchmarks to dynamic behavioral profiling, adopting standardized methodologies, regulatory certainties and ethical guidelines should be prioritized. We encourage more research in these directions to bridge the gap between current constrained NLG evaluations and the model performance in real-world settings.

## 8 Discussion

In what follows we summarize the main challenges for constrained NLG and outline open problems, then we present the most promising research directions in the authors' opinion for advancing the state-of-the-art for safe and reliable constrained NLG.

### 8.1 Open Challenges

In our view, constrained text generation is a more difficult problem compared to other instances of text generation. The difficulty arises from a multitude of factors, including lack of model expressiveness which makes

it difficult for current models to incorporate constraints into the objective function, lack of suitable evaluation metrics to assess the extent to which constraints are satisfied (which becomes even more challenging in the presence of multiple constraints), difficulty in the constrained optimization of non-differentiable reward functions, and finally lack of constrained text generation datasets that are illustrative of a wide diversity of constraints. Due to these pressing issues, constrained text generation remains an open challenge in the research community. Advancing the state-of-the-art requires considerable collective and focused effort.

**Multiple constraint satisfaction** Most approaches proposed for constrained text satisfaction focus on generating sentences that meet one single desired constraint, nevertheless generating sequences that simultaneously satisfy multiple lexical constraints is an important open research problem in text generative models (Liu et al., 2019a; Latif et al., 2020; Hsieh et al., 2021). While incorporating one constraint is already hard enough due to lack of model expressiveness, incorporating multiple constraints poses significant challenges in terms of defining the loss function accounting for all the desired constraints, difficulty in optimizing it and evaluating whether each constraint is satisfied. Approaches that convert the multiple constraint satisfaction problem into allowing the inclusion of pre-specified lexical constraints at decoding time are not optimal either: on the one hand, decoding complexity increases exponentially or linearly in the number of constraints, and on the other hand forcing constraints at every step of the generation process impacts the quality and naturalness of generated texts (Post & Vilar, 2018). Moreover, many model architectures are designed for sequential sentence generation only (vs. non-monotonic text generation) and it is non-trivial to impose decoding time constraints while maintaining optimal text generation quality (Miao et al., 2019).

Prompting methods are used to evaluate to what extent multiple constraints are satisfied by instruction-tuned LLM models (Jiang et al., 2024). Single constraints are added to instructions in an incremental fashion, allowing to estimate the upper limit of instruction following capabilities in LLMs and assess the difficulty level at which models fail to follow instructions with multiple constraints. Overall, the more constraints are added to an instruction, the more rapid the decrease in performance of state-of-the-art LLM models; on average, at most three constraints are satisfied. Constraints such as role-playing, reasoning in complex situations, numerical planning, suggestion generation, recognizing and following patterns are identified as the most difficult constraints to satisfy. Prompting with multiple, verifiable, fine-grained constraints is proposed for assessing discrepancies/misunderstandings in following instructions that lead to unintended outputs (Zhou et al., 2023a; Sun et al., 2023a). Nevertheless, making constrained NLG evaluations relevant to real-world applications is crucial for improving the reliability of conclusions drawn from current benchmark evaluations.

**Dynamically defined constraints** Current approaches to constrained text generation assume there is prior knowledge of the constrained textual attributes and the finite set of values these attributes can take. Nevertheless, there are situations when it may be desirable to impose constraints dynamically, for eg. in conversational systems depending on the system user's statements, reactions and emotions. When dynamically defining constraints, the main challenges are the lack of model expressiveness and robust ways to evaluate whether these constraints are satisfied. In the literature, controling the realization of a sentence based on another's sentence syntax and semantics is a less explored setting for constrained text generation with dynamic constraints which does not require prior knowledge of all the values the control variable might take on (Chen et al., 2019). Disentangled latent space representations of syntax and semantics are essential for the manipulation sentence attributes in tasks such as unsupervised paraphrase generation and syntax-transfer generation (Bao et al., 2019). In the context of LLM models, handling dynamic and complex application constraints remains challenging and relatively under-explored area of research. Commonly used solutions for incorporating dynamic constraints leverage the reasoning and planing capabilities of LLM models via strategies such as model fine-tuning and reflection-based reasoning, for example Chain–of-Thought (Wei et al., 2022), Tree-of-Thoughts (Yao et al., 2024) or Self-Play fine-tuning (Chen et al., 2024b). Nevertheless, these approaches typically address constraints on a case-by-case basis, limiting their generalizability. Open questions are how to represent constraints for LLMs effectively, guide them to reason within those constraints, and accurately assess the correctness or fallacies in their reasoning process (Wei et al., 2024).

**Generative reasoning** Current large-scale text generation models display impressive ability to generate fluent texts, nevertheless composing realistically plausible sentences in the presence of constraints remains a significant open challenge. This is illustrative of all challenges associated with constrained text generation, including lack of model expressiveness, lack of suitable evaluation metrics, difficulty in constrained opti-

mization and lack of constrained text generation datasets. Endowing generative models with commonsense reasoning abilities is an important milestone towards advancing machine understanding and intelligence. In general, the great majority of models proposed in the literature only exploit superficial cues via self-attention to solve NLP tasks, without relying on syntactic information or complex reasoning (Pham et al., 2021).

LLM models trained with reinforcement learning can perform complex reasoning tasks by thinking through a problem, producing a series of intermediate reasoning steps that allow to recognize and correct mistakes before attempting to give the final answer. Prompt-based reasoning with LLMs has lead to rapid advancements on many constrained NLG tasks, including mathematical, logical and commonsense reasoning problems, code generation, question answering, text summarization, machine translation, etc. (Plaat et al., 2024; Liu et al., 2025a). Despite the strong performance of LLMs on certain reasoning tasks, the extent to which LLMs are actually capable of reasoning and handling complex deductive problems (vs. exploiting superficial cues and shallow patterns in the data) remains uncertain (Huang & Chang, 2023; Hosseini et al., 2024). Compared to human performance, LLMs multi-step logical and commonsense reasoning performance is lagging behind humans in various real-world domains such as murder mysteries, object placement or team assignment (Sprague et al., 2024). Significant decline in LLM reasoning performance is reported as the complexity of constraint satisfaction problems increases, a phenomenon referred to as "the curse of complexity for reasoning" (Lin et al., 2025). Scaling up the number of reasoning tokens generated during inference, and ensuring the correctness and verifiability of the reasoning chain may alleviate the issue to some extent.

**Attribute specific datasets** The lack of annotated datasets for attribute specific text generation constitutes a bottleneck in the development and adaptation of models for tasks that require fine-grained control over style and topics. For example, in dialogue systems the absence of attribute annotated conversational datasets that can be used for fine-tuning large scale pre-trained models limits control over the generated responses for a desired attribute (Madotto et al., 2020). Moreover, such attribute annotated datasets can help with the personalization of dialogue systems, make dialogues safe, supportive and engaging (Serban et al., 2015; Zhang et al., 2018a; Ge et al., 2024). Personalized dialogue agents that display consistent personalities and viewpoints overcome the unsatisfying experience of a persona-free chit-chat model and empower practical applications such as personalized conversations. Nevertheless, imposing conversational goals on a dialogue agent for learning target-guided strategies requires keyword-augmented conversation datasets for learning how to steer the conversation towards a designated target subject (Tang et al., 2019).

**Rule constraints** While most research that is currently trying to address constrained text generation is focusing on the incorporation of pre-defined utility or lexical constraints to various degrees of success on simple tasks with narrow scope (Ashok & Poczos, 2024; Sun et al., 2023a; Zhang et al., 2023), the satisfaction of rule based constraints is equally relevant, particularly when used to define format and syntactic conditions on the output. However, the lack of model expressiveness makes it challenging to incorporate rule based constraints into the loss function at training time. We encourage more effort in this direction likely to open a plethora of new possibilities in how constraints are specified, incorporated and satisfied in models particularly designed for constrained neural text generation.

Prompt-based control for lexically constrained text generation remains challenging for the following reasons: *i)* current LLM models tend to display *position bias* (Liu et al., 2024c), and only satisfy lexical constraints that appear within specific positions in the input, *ii) decoding parameters lack of sensitivity* to incorporate lexical constraints, and *iii) complexity of compound word constraints*, which are often misinterpreted or altered in meaning by LLM models. Overall, LLMs struggle to adapt to increasingly complex lexical constraints with prompt-based control (Li et al., 2024a); the more lexical constraints are added to the prompt, the more significant the decrease in performance of LLM models for constrained text generation (Jiang et al., 2024). In addition, there is an inherent trade-off between generating text of high quality and satisfying hard constraints (Iso, 2024). LLMs often struggle to meet fine-grained hard constraints (Sun et al., 2023a), and evaluations on out-of-distribution constraints aiming to differentiate constraint-following abilities from over-fitting (IFEval (Zhou et al., 2023a) vs. IFEval-OOD (Lambert et al., 2024)) report that a lot of the claimed success of prompts targeted for constrained instruction following may be simply attributed to overfitting.

**Evaluation of constrained text generation** In general, evaluation of text generative models is an open challenge. The field is missing robust automated evaluation metrics that correlate with human judgements

across multiple dimensions of text quality. Evaluation of models for constrained text generation is currently done using the same flawed existing metrics commonly used in unconditional and conditional text generation evaluation, or in an informal way often times in the absence of a rigorous evaluation procedure. Human evaluation remains the gold standard way to assess text quality, however designing evaluation metrics tailored specifically at assessing whether generated texts meet desired constraints altogether with new benchmark datasets for the evaluation of constrained text generation are important next steps (Latif et al., 2020; Ruan et al., 2024; Chen & Wan, 2023; Zhou et al., 2022a; Hu et al., 2024).

**Adversarial Attacks** Adversarial examples exploit vulnerabilities in text generation models and represent an active research area. Adversarial triggers in the form of input-agnostic sequences of tokens concatenated to any input dataset can trigger a pre-trained language models to produce biased, racist and discriminatory outputs even when these models are carefully fine-tuned and optimized against adversarial triggers (Wallace et al., 2019). Gradient-based adversarial trigger phrase search techniques are used to generate input prompts to a pre-language model that induce biases in the generated output and allows to study strategies for bias mitigation (Sheng et al., 2020). Constrained text generation models that are robust to adversarial attacks are needed for the beneficial use of machine learning and artificial intelligence technology in real world applications, as well as to mitigate any potential societal harms and biases associated with the deployment of large language models (Chowdhury et al., 2024; Gallegos et al., 2024).

Other important open challenges include the use of constrained text generation for personalized agents in a wide variety of contexts (Zhang et al., 2024b; Liu et al., 2025b), such as in dialogue settings (Zhang et al., 2018a), and new benchmark datasets that are reflective of real-world constraints for both training/fine-tuning and evaluating constrained text generation models (Ziyu et al., 2023; McIntosh et al., 2024; Xu et al., 2024).

### 8.2 Promising Research Directions for Advancing Constrained NLG

Despite the many open challenges, we believe there are promising approaches in the literature that merit special attention for advancing constrained NLG. We present these below.

**Reinforcement Learning from Human Feedback (RLHF)** is the predominant paradigm for aligning LLMs to human preferences given helpfulness, harmless and safety constraints (Ouyang et al., 2022). RLHF performance is strongly dependent upon the quality of the reward model, and defining rewards for real-world tasks, especially with the presence of constraints, is non-trivial. Reward functions are fragile and notoriously difficult to specify, particularly for tasks with complex goals (McKinney et al., 2023). An outstanding RLHF challenge is the issue of reward hacking, where LLM policies learn to exploit failures of the reward model and achieve seemingly high rewards without meeting the underlying objectives (Rame et al., 2024b).

Despite these challenges, RLHF can play a key role in advancing constrained NLG, conditioned on the design of more robust and reliable reward functions. To mitigate reward hacking in particular, it is necessary to have reward models that can reliably score generations despite distribution shifts, are robust to label noise and inconsistencies in human preferences. Preliminary approaches that explore prediction ensembling (Christiano et al., 2017) or weight averaging (Rame et al., 2024b) of multiple reward models are designed to act as regularization preserving only those mechanisms that are invariant across runs, helping reduce reliance on spurious features and memorization of corrupt/noisy training examples. While these approaches may help delay reward hacking to some extent, they do not fully solve the problems of reliability under distribution shifts and robustness to noisy labels. We believe there is a lot of space to explore more efficient solutions for training robust reward models for real-world constraints that accurately reflect human preferences, entirely prevent reward hacking (instead of just delaying it), and generalize to out-of-distribution settings.

RL policies that perform well for diverse reward functions (not just one reward model) can accommodate diverse user preferences and advance NLG with multiple constraints. Multi-objective reinforcement learning (MORL) algorithms can be used to learn Pareto-optimal policies and control the learnt policies accounting for multiple objectives (Liang et al., 2024). For example, multi-objective reward modeling allows to dynamically control the trade-off between diverse user preferences via arithmetic operations in the vector reward space (Wang et al., 2024a). However, in many real-world settings it may be difficult to accurately specify constraints mathematically; in such situations, it may be possible to learn constraints directly from user provided demonstrations, even when the reward function is unknown (Lindner et al., 2024; Malik et al., 2021).

**Mechanistic Interpretability** Recent works aiming to understand the inner workings of LLMs find that these models have internal representations that encode concepts in a disentangled manner. If one can identify which part of the representation subspace corresponds to a given concept, then it its possible manipulate the concepts expressed by the model through algebraic manipulation of the representation (Wang et al., 2024c). The linear representation hypothesis (Park et al., 2024a) posits that high-level concepts are represented linearly as directions in the representation space. Assuming it is possible to identify these linear concept representations, linear algebraic operations can be performed on the representation space for fine-grained control of LLM outputs. Interventions using steering vectors that change the value a concept takes without changing other concepts show that in carefully designed test cases it is possible to change, for eg. the output from English to French by adding a suitable English/French steering vector. If the linear representation hypothesis were to hold true, this could potentially open up new methods that advance constrained NLG. Open problems are the identifiability of learned representations, to what extent they capture real-world structure, and what assumptions need to be made about the geometry of the representation space.

**Causal Interventions / Causal Probing / Inference Time Interventions** Recent work shows that it is possible to perform direct manipulation of computational mechanisms inside LLMs. In particular, causal tracing approaches first identify neuron activations corresponding to a particular concept, then edit the corresponding weights to change model outputs in a desirable way (Meng et al., 2022; Meng et al.). However, the connection between causality-based localization and model editing is still unclear, as localization performed by causal tracing is not indicative of which layer to select for model editing (Hase et al., 2024). Developing more reliable causal mechanisms for localizing where knowledge is stored inside a neural network, as well as robust ways to edit the internal knowledge of LLM models to ensure generated outputs satisfy given constraints, can pave the way for advancing constrained NLG.

Causal probing can be used to analyze how intervening on latent properties of the model's representation have an impact on the model outputs (Canby et al., 2024). For example, linear probing has been used to predict whether the answer will be truthful or not before it is actually generated (Joshi et al., 2024). While measuring the effectiveness of probing interventions in LLM models is an open research area, causal probing is a promising direction to explore whether constraints are present in the generated output before generation even begins. Inference-time intervention (ITI) (Li et al., 2024c) locates directions that correspond to specific concepts (for eg., truthfulness) and shifts model activations along these directions at inference time. Potentially integrating ITI with causal probing and intervention mechanisms, and a better understanding of the geometry of the representation space, could help enforce complex attributes/constraints in the output.

**Reasoning-based approaches / Constrained Decoding approaches / Rejection Sampling** Graph-constrained reasoning (Luo et al., 2024) aims to connect the unstructured reasoning in LLMs with the structured knowledge found in knowledge graphs (KG); the model constrains the LLM decoding process to reasoning paths that encode KG information. Generating KG-grounded paths helps alleviate reasoning issues due to lack of knowledge, mitigates hallucinations and enhances the faithfulness of generated responses. Constrained chains of reasoning (Lin et al., 2024b) leverage domain knowledge and the causal relations between concepts to construct reasoning chains that improve consistency of the generated responses.

Constrained decoding approaches enforce adherence to constraints during generation, while (ideally) minimally intervening on other non-target aspects to avoid misalignment (Beurer-Kellner et al., 2024). Guided decoding (Lu et al., 2021) employs an auxiliary evaluation function that captures to what extent partial outputs satisfy given goals; the method can be combined with search algorithms to generate outputs that satisfy specific constraints. Combining inference-time search algorithms such as Monte-Carlo Tree Search (MCTS) with RLHF fine-tuned models (Liu et al., 2024a) demonstrates that value-guided decoding with MCTS is a crucial component for achieving model steerability and constraint satisfaction (for eg., sentiment steering, toxicity reduction, helpful and harmless chatbots). Value models trained as byproducts when aligning LLMs to human preferences have only recently been employed as evaluation functions for scoring partial/incomplete sequences and steering LLM models; it would be interesting to use MCTS as a policy optimization operator to search for contrained sequences with high rewards. Overall, we believe that combining reasoning-based approaches with guided decoding and MCTS has a lot of potential to improve the state-of-the-art for constrained NLG, including more faithful, consistent generations and less hallucinations.

Statistical rejection sampling can be used to discard partial samples that do not meet given constraints; this technique has been widely employed in LLAMA-3 family of models (Dubey et al., 2024), and is found to improve the alignment with human preferences in constrained optimization settings (Liu et al., 2024d). Best-of-$N$ rejection sampling (Stiennon et al., 2020) draws $n$ samples from the LLM, ranks them on the target attribute of interest, and returns the best sample. Despite its simplicity, this strategy has been found to be surprisingly effective in practice, however Best-of-$N$ sampling comes at considerable inference cost. Approaches trained to mimic this distribution achieve high win-rates while minimally affecting other off-target aspects of the generation (Gui et al., 2024), allowing for better control of LLM models.

**Improving model architectures** The mismatch between how LLM models are trained with a next-word prediction objective and how they are used in practice (for eg., for long-term open-ended dialogue with users) leads to inconsistency in their behaviour over long horizons. The Transformer attention mechanism decays over long exchanges, causing chatbots to stray away from prompted behaviour and resulting in instruction drift that degrades the quality of the outputs over lengthy dialogues (Li et al.). Improving instruction stability in LLMs, particularly in long-form conversations, can lead to more stable and robust prompting, improve the performance of current models and their abilities to generate texts accounting for given constraints. There is a need for better understanding how LLMs use input context and potentially come up with novel designs for attention mechanisms that more robustly capture information within long input contexts.

Autoregressive LLMs fail to generalize in surprising ways. The reversal curse (Berglund et al., 2024) is one instance where a model trained on a sentence of the form "A is B" will not automatically generalize to the reverse direction "B is A", therefore failing to deduce the reverse relationship – this directly impacts the LLM models' ability to generate constrained texts. Further analysis of LLMs generalization using influence functions (Grosse et al., 2023) finds that training examples that match the order ("A precedes B") are far more influential than examples with reverse order ("B precedes A"). Fine-tuning and data augmentation approaches are used to alleviate the issue, however it does point to a basic failure of logical deduction in LLMs training. Incorporating logical deduction and causal-based reasoning during LLM training could help.

**Accounting for constraints / Better guidance during training** Classifier guidance (Dhariwal & Nichol, 2021) introduces an extra trained classifier to guide diffusion model generations in particular desirable directions. Methods aiming to improve this approach propose classifier-free guidance (Ho & Salimans, 2022), showing it is possible to steer LLMs using a pure generative model; a conditional and an unconditional diffusion model are jointly trained, and their score estimates are combined to achieve fine-grained control over the generated outputs. There is a lot of potential to extend the use of such approaches beyond the vision domain to the text domain for more robust constrained NLG.

**Multi-objective / pluralistic benchmarks** Aligning LLM models to pluralistic human values requires the capability to accurately steer models in directions representing a diverse set of human values and perspectives (Sorensen et al., 2024). Steerable pluralism, i.e. faithfully steering LLMs to represent particular attributes or perspectives, plays a key role in personalizing and customizing models to various users or target populations. Nevertheless, current benchmarks for constrained NLG are monistic, focusing on a single objective. Pluralistic benchmarks have more than one objective to maximize (each objective is measured separately) and allow for more explicit trade-offs between constraints at inference time.

**Hybrid Human-AI Collaborative Approaches** Interactive systems that allow LLM models and their end users to write collaboratively have the potential to enhance constrained text generation by empowering users to choose which (partial) model generations are in line with their needs and expectations. CoAuthor (Lee et al., 2022) proposes a collaborative human-AI approach for text generation where a writer and a model take turns interactively in writing a story and editing it. Such methods empower users with fine-grained control over the model outputs and open up new opportunities in assistive writing in a steerable fashion. Causal inference can play a key role in modeling the human-AI collaboration byanswering counterfactual "what-if" questions on how the outcome of the collaboration would change if humans employed a different text editing/refinement strategy. For example, causal estimands (Incremental Stylistic Effect) (Zhang et al., 2024a) can be used to measure the average impact of infinitesimally shifting a text towards a specific style, such as increasing the degree of politeness or formality.

## 9 Conclusion

In this work, we have presented the reasons why constrained natural language generation is an important, yet highly challenging and largely unsolved research problem. Our first contribution consists in clarifying the difference between the ambiguous use of unconditional, conditional and constrained terms in the natural language generation literature, and draw clear boundaries between these concepts by exemplifying instances of natural language generation tasks with their associated conditions and constraints. Among different paradigms of text generation, we consider constrained text generation to be particularly challenging (if not the most challenging), yet also extremely useful. We identify general reasons why constrained natural language generation deserves significant more attention in the research community, including the lack of model expressiveness in incorporating constraints into the objective function at training time, difficulty in constrained optimization algorithms, the lack of suitable evaluation metrics for robustly assessing, comparing model outputs and claiming success in constrained natural language generation, as well as the lack of constrained text generation datasets/benchmarks that are representative of a wide range of real-world constraints for training, fine-tuning and evaluating these models. We then survey a representative body of recent literature on constrained text generation using neural networks, presenting the main approaches and methods used, as well as their limitations. Our work serves as an informative guide for both researchers and practitioners to become familiar with the current methodology and main challenges, in the hope of advancing state-of-the-art constrained NLG. We invite future work in solving the outlined challenges for better, useful, safer and more robust constrained natural language generation and evaluation.

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
