# OpenReview forum: "Why is constrained neural language generation particularly challenging?"
_TMLR — Accepted by TMLR_

### Review · Reviewer_Umg2 · 2024-10-16

**Summary Of Contributions:**

The following contribution provides an extensive review of natural language generation, delving into the various constraints, metrics, and conditions that critically influence the quality of generation and the evaluation processes of current neural language models. This detailed analysis aims to shed light on the key factors that drive the performance and effectiveness of these models in real-world applications, highlighting both the advancements and the ongoing challenges faced by researchers and practitioners in the field.

Generally, the contribution is well-formed and has a clear and concise structure. However, some unclear points need to be examined thoroughly.

**Audience:**

Yes

**Broader Impact Concerns:**

The impact is not decisional. Although the paper is interesting and deals with important topics, it needs updating.

**Claims And Evidence:**

No

**Requested Changes:**

The contribution proposes a multidimensional survey of the NLP world. Although it was well written and organized, there are points that make the contribution of little value.
In particular:

- The introduction is clear, but research questions or bullet points should be used to clearly understand the direction of the work;
- The citations are truly archaic; it is not possible to present a work without citations from at least the year preceding the submission
It may be important to make a hierarchical map of the works (which still need to be reviewed).

The contribution has a good structure but definitely needs updating to be competitive and in a good form.

**Strengths And Weaknesses:**

It is a very interesting topic covered from a compilation point of view (no experiments).

The writing is clear and understandable; however, there are points that need to be reviewed urgently.

---

> ### Author Response · Authors · 2024-12-15
>
> Thank you for your review!
>
> We will add bullet points in the introduction to clarify the direction of our work. We will also include additional relevant papers that have appeared over the past year - please note that our survey already discusses relevant recent constrained NLG papers (citations may show the date when these works were posted to ArXiv, however many of these papers have been officially published in conferences over the past year - we will update the bibliography for these).
>
> Our goal is to provide a comprehensive and multidimensional overview of the constrained NLG field, and reviewer ndnQ has appreciated our extensive focus discussing methodology dating from both pre and post LLM era. We have included the representative citations that are essential for describing the general methods, evaluation, and applications in our survey.  Given the fluidity of the literature, we can't enumerate all special cases and methods of constrained NLG. Our goal is to provide a meaningful and compact roadmap of the research direction for the readers rather than an exhaustive list of recent papers. Please refer to our response to reviewer ndnQ for additional recent papers relevant to constrained NLG.

---

### Review · Reviewer_r3Cn · 2024-11-27

**Summary Of Contributions:**

This work provides an extensive survey on constrained neural language generation (NLG), focusing on formally defining the challenges, methodologies, and applications of imposing constraints on output text. It categorizes NLG tasks into generic, conditional, and constrained paradigms and highlights the distinction between conditions (testable statements about the input) and constraints (testable statements about the output). The work comprehensively reviews many approaches (decoding, fine-tuning, discriminative, edit- based, and architecture adaptation) and the challenges associated with constrained NLG, such as lack of model expressiveness, suitable evaluation metrics and datasets.

Summary of Contributions:
1. Definition and Categorization: The paper provides a clear distinction between conditions and constraints, offering a structured framework for understanding constrained NLG.
2. Extensive Survey: It consolidates a broad range of methods for constrained text generation, detailing their strengths and limitations.
3. Highlighting Challenges: This work identifies key barriers to advancing constrained NLG, including the non-differentiability of constraints and limited datasets.
4. Applications and Metrics: Authors in this survey outline real-world applications, such as dialogue systems, machine translation, and summarization, and discusses existing evaluation metrics for constrained NLG.

**Audience:**

Yes

**Broader Impact Concerns:**

1. Bias and Safety: The survey mentions mitigating toxicity and bias in generated content
but does not delve into ethical concerns of deploying constrained NLG systems.
2. Dual-Use Risks: Constrained NLG can be misused to generate persuasive but harmful content, such as disinformation or propaganda.
3. Societal Applications: The broader implications of constrained NLG in sensitive domains like healthcare or legal advice need more exploration.

Comments, Suggestions, and Typos:

1. Comments:
Clarify the distinction between soft and hard constraints with examples. Expand on how constraints can be dynamically adjusted during generation. Provide a visual taxonomy of methods and applications for constrained NLG.

2. Suggestions:
Use tables or charts to summarize approaches, applications, and challenges for better readability.
Include a dedicated section for "Future Directions" to organize actionable insights more effectively.
   3. Typos and Formatting Issues:
Page 3: "generative commonsense reasoning with a constrained text generation task is more challenging for machine learning models" — rephrase for clarity.
Page 7: "stylistic constraints are immediately relevant to the task of text style transfer" — redundant phrasing, streamline.
Several sections use "eg.," without periods; standardize to "e.g.," for consistency.

**Claims And Evidence:**

Yes

**Requested Changes:**

1. Propose New Metrics: Include detailed discussions on potential novel evaluation metrics for constrained NLG tasks.
2. Expand Dataset Review: Provide a categorized list of existing datasets and specific gaps that need to be addressed.
3. Highlight Positive Trends: Balance the discussion of challenges with examples of successful constrained NLG applications and breakthroughs.
4. Suggest Research Directions: Offer concrete, actionable suggestions for addressing identified challenges, such as integrating reinforcement learning with constraints or developing hybrid architectures.

**Strengths And Weaknesses:**

1. Insufficient Novelty: The paper is primarily a survey and does not propose new methods, datasets, or evaluation frameworks.
2. Limited Practical Insights: While challenges are listed, actionable strategies for overcoming these limitations are underexplored.
3. Superficial Metrics Discussion: The analysis of evaluation metrics for constrained NLG lacks depth and does not suggest comprehensive solutions.
4. Dataset Diversity: Although dataset limitations are acknowledged, the survey does not provide a detailed catalog of available datasets or gaps.
5. Heavy Focus on Challenges: The paper dedicates significant space to challenges without sufficiently emphasizing progress or success stories in the field.

---

> ### Author Response · Authors · 2024-12-15
>
> Thank you for reviewing our paper! We would like to address the concerns you raised:
>
> 1. Insufficient Novelty: The paper is primarily a survey and does not propose new methods, datasets, or evaluation frameworks.
>
> You are correctly pointing out that our paper is a survey and it has indeed been submitted to the survey track of the TMLR journal. The goal of a survey is to present the latest trends and advances in a field, critically analyze where the field is in the present, what are the current limitations, as well as highlight open problems and promising future work directions (i.e. inform the reader on what has been done, where the field is now, and what needs to be done to address current gaps and limitations to further advance the field). We believe our present work covers all these aspects, and contributes with important insights that educate and inform any reader interested in the topic of constrained NLG. We thoroughly review existing methods, datasets and evaluation frameworks, and provide an informative guide for any researcher interested in the topic of constrained NLG. Proposing new methods, datasets and metrics is beyond the scope of our survey.
>
> 2. Limited Practical Insights: While challenges are listed, actionable strategies for overcoming these limitations are underexplored.
>
> Thank you for pointing this out. Please see our response to reviewer ndnQ for a detailed discussion of practical insights and actionable strategies we recommend for advancing constrained NLG. We will include these insights in our updated manuscript and we will strengthen the discussion in Section 8 of the paper with actionable strategies and concrete recommendations for overcoming the limitations outlined.
>
> 3. Superficial Metrics Discussion: The analysis of evaluation metrics for constrained NLG lacks depth and does not suggest comprehensive solutions.
>
> Current evaluation metrics used for constrained NLG largely overlap with well-established evaluation metrics in the NLP literature. Unsurprisingly, there are currently only a few automated evaluation metrics particularly designed for constrained NLG evaluation (which we outline and categorize in Section 6). A large majority of constrained NLG papers have adopted evaluation measures that are largely applicable to text evaluation at large, which is one of the reasons limiting the progress of the field.
>
> In our response to reviewer ndnQ, we are discussing limitations of current evaluation paradigms and metrics, including test set contamination of current evaluation benchmarks [35], lack of holistic evaluations and standardized evaluation guidelines [38], the researcher’s choice of evaluation metric falsely giving the impression of progress (i.e. nonlinear or discontinuous metrics produce apparent emergent abilities, whereas linear or continuous metrics produce smooth, continuous predictable changes in model performance [36]), summarizing performance as a single scalar value offers an incomplete picture of the model performance and can lead to reward-hacking [33]. In general, any evaluation metric for constrained NLG should follow the design principles for trustworthy evaluation and be relevant, robust, optimization friendly and indicative of downstream performance. The purpose of this survey is to outline the current state of the constrained NLG field and not to find novel and comprehensive solutions.
>
> 4. Dataset Diversity: Although dataset limitations are acknowledged, the survey does not provide a detailed catalog of available datasets or gaps.
>
> Thank you for pointing this out, we will discuss available constrained NLG datasets and their limitations in more detail. As the field is rapidly evolving, it is likely we may have missed some of the newest benchmarks proposed for constrained NLG. We will include these along with a discussion of gaps in the literature, for eg., types of constraints not currently covered by current benchmarks.
>
> 5. Heavy Focus on Challenges: The paper dedicates significant space to challenges without sufficiently emphasizing progress or success stories in the field.
>
> Our survey covers a wide diversity of methods that have successfully contributed to advancing the field of constrained NLG in recent years (although no method is perfect and there are still gaps and open problems left to address which we cover in detail). Our goal is to highlight that despite the progress that has been made, constrained NLG remains a significant challenge for reasons the research community has overlooked. Although there is still more research needed to advance constrained NLG,  we agree we could better emphasize the successes so far.

---

> > ### Author Response · Authors · 2024-12-15
> >
> > Requested Changes:
> >
> > Thank you for your feedback and suggestions! We will incorporate your requested changes in our updated draft:
> > - expand the discussion section with promising approaches for advancing constrained NLG;
> > - include a detailed overview of the reasons behind the mismatch between constrained NLG benchmarks and real use case scenarios, add additional benchmarks and discuss gaps such as types of constraints not currently covered by existing benchmarks;
> > - highlight successes of current models and discuss positive trends in constrained NLG;
> > - suggest promising research directions for more robust and reliable constrained text generation.
> >
> > For more details regarding the points above please see our response to reviewer ndnQ.
> >
> >
> > Broader Impact Concerns:
> >
> > These comments apply to NLG in general, not particularly for constrained NLG (i.e., they apply to conditional NLG as well). We will mention in our updated draft that a particular purpose of developing constrained NLG is to constrain the NLG models from generating these harms - and as a special case of constrained NLG, this practice also suffers from all the challenges we described.
> >
> > In the survey we are often highlighting how constrained NLG can be used for finer-grained control to mitigate against biases and toxic outputs of generative models, prevent societal harms and ensure fairness to demographic groups.  We will further emphasize these points.
> >
> > Comments and Suggestions:
> >
> > Thank you for your comments and suggestions! We are glad to address them in our updated draft!
> >
> > Typos: Fixed, thank you!

---

### Review · Reviewer_ndnQ · 2024-12-02

**Summary Of Contributions:**

This paper provides a survey on **constrained neural language generation**. It begins with a definition off constrained neural language generation, and highlights its important difference with the **conditional** language generation problem. Concretely, the former is defined as imposing constraints on the output of the model (e.g. the model needs to follow a particular style or constraint), whereas the former is defined as conducting neural language generation, conditional on some input (e.g. the input condition can be a sentence in the source language, in the case of neural machine translation). The paper then proceeds to give examples of constrained NLG tasks, including the typical input condition & output constraints for each task, such as format constraints, syntactic constraints, semantic constraints, etc.

After defining the problem and the typical types of constraints & conditions for each task, the paper then surveys prior approaches on constraint NLG, including decoding approaches, fine-tuning approaches, discriminative approaches, edit-based approaches, adapting existing models & architectures to accommodate the constraints, and also prompting-based approaches for LLMs. The paper then outlines current evaluation metrics, benchmarks, and datasets for constrained NLG evaluation, and highlights why this remains a challenging open problem in the field (e.g. noise in automatic metrics, lack of standardized procedure for human evaluation, etc.).  The paper then summarizes the key open challenges in constrained NLG, including the key problems that the field should tackle in order to make progress on more realistic constrained NLG setups (e.g. satisfying multiple different constraints at once, rather than focusing mostly on one type of constraint, as is done in most constrained NLG evaluation datasets today).

**Audience:**

Yes

**Broader Impact Concerns:**

No broader impact concerns come to mind.

**Claims And Evidence:**

Yes

**Requested Changes:**

1. **Critical**: Refactor parts of the paper to be made more concise (Weakness point 2).
2. **Critical**: Improve the notations (Weakness point 1).
3. **Critical**: Expand the discussion on: (i) what promising approaches we should try more for constraint NLG, given that the problem is very much a challenging one & will likely require more modelling / architectural innovations; (ii) what mismatch there is between current constrained NLG evaluation metric, datasets, and benchmarks, in contrast to real-world applications of the problem (this will highlight what kinds of datasets & metrics we should develop next as a community); and (iii) what the areas & approaches we should revisit are in the LLM era (e.g. are there any techniques that we should combine with recent LLM progress, and why). (Weakness point 3).
4. **Important**: Clarify the discussion with respect to the perplexity point, and how it is potentially comparable across different models (Weakness point 4).
5. **Important**: Improve the presentation of the paper & fix the typos (Weakness point 5).

**Strengths And Weaknesses:**

# Strengths

1. The paper provides a comprehensive coverage of prior work on multiple axes relating to constrained NLG, including different types of approaches to this problem & their limitations, datasets & metrics, etc. I appreciate the fact that the surveyed approaches encompass many different time periods, from the pre-LLM era & the current set of more recent work, including LLM prompting, etc.

2. I like the fact that the paper dedicates a section on evaluation metrics & datasets and benchmarks, and highlights how these factors constitute a key bottleneck to constrained NLG today (in particular, the lack of a reliable automated evaluation metric, the lack of a standardized protocol for conducting human evaluation, and the fact that current constrained NLG datasets are not necessarily representative of how constrained NLG is done in real-world applications).

# Weaknesses

1. Some of the notations used in the paper are confusing / inaccurate. For example, the negative log likelihood objective at the top of page 3 has a summation only over the different **sentences** in the dataset, where each sentence is indexed by $k$. However, in standard, autoregressive language models, we also need to do a summation over the different **token positions** in the document, which is currently neglected in the current notation. The correct language modelling objective is: $\ell(D) = \sum_{k=1}^{|D|} \sum_{i=1}^{N_k} \log p_{\theta}(x_{i}^{k} \mid x_{<i}^{k})$, where $N_k$ denotes the length of the $k$-th sentence / document in the dataset $D$. The same point also holds for the equation in Section 2.2, and also Section 2.3 (top of page 4). The notation also does not exactly follow standard convention of denoting vectors / sequences of tokens with bold (e.g. $\boldsymbol{x}_{<i}^k$ is denoted in bold because it denotes all the previous tokens at position $<i$), and individual token with non-bold (e.g. $x_i^k$ is denoted in bold because it is a single token). Also the model parameters $\boldsymbol{\theta}$ should ideally be denoted in b old, because it refers to the full set of model parameters, rather than a single scalar.

2. I find some sections of the paper to be rather belaboured, and does not add too much value to the overall theme of the paper. For example, section 4 provides an overview of constrained NLG tasks like machine translation, dialogue systems, etc. I do not find this section to be necessary, because (i) many TMLR readers will already be broadly familiar with the tasks that are discussed here; and (ii) the types of NLG constraints for each task are already well-summarized in Table 1, rendering a comprehensive description in the text unnecessary. Another similar section that can be made clearer & more concise is the evaluation section. Page 22 of the paper outlines the limitations of current evaluation metrics, including word overlap metrics like BLEU, etc. This part reads to me like a rather generic overview of these automated metric's limitations, which are already quite well-known in the field. A lot of these limitations are generic, and therefore not particularly relevant to the problem at hand (constrained NLG).

3. There are a few interesting points that this paper raised that warrants further discussion in the paper, which are still missing from the current version.
    - The first point relates to the promising **approaches** for constrained NLG. There is already a lot of discussion on the paper about what is still missing from the setup & current work & datasets & metrics, but an equally important angle is: What approaches should the field try in order to get better results in this problem. Given that: (i) the evaluation metric is noisy & hard to hill-climb on; (ii) accurately measuring our progress is markedly difficult; and (iii) real-world constrained NLG applications will be **even more challenging** than what is reflected in the current datasets of today (e.g. real-world applications will require fulfilling multiple different constraints on noisy datasets), what do the authors think as the main way forward? Should we hold off on coming up with new models until we come up with better evaluations? Or are there approaches that the field is not trying enough of? Should we invest more in e.g. diffusion models? Or something else?
    - The paper highlights the mismatch between the constrained NLG evaluation benchmarks & real-world use cases, which in my opinion is a great point and a timely problem. However, the paper does not dive into more detail on the key points of this mismatch. What kinds of mismatch are there? And based on this, what kinds of datasets & metrics should we develop next in order to more accurately measure progress in this direction? Should we focus on problems with multiple constraints? Or should we focus on more realistic (e.g. real-world LLM use cases) & diverse domains for evaluations? Or something else?
    - What are the areas that we should revisit? Is there any prior work from the pre-LLM era that we should revisit, because they are likely to combine nicely with recent LLM advances? It would be great if this survey paper does not solely focus on what has been done and what is still missing, but also take a stance on the most promising ways forward.

4. I find the assertion in Section 6, "Perplexity" paragraph, page 20 to be dubious. In particular, it states that "... how likely a sentence is generated by a given model is not directly comparable across different models." I would argue to the contrary, that one can, indeed, compare perplexity from different models (we have been doing it since the days of the Penn Treebank language modelling evaluation). The main thing is that we need to divide the total negative log likelihood from each model with the same denominator (typically the number of tokens or characters or bytes in the whole evaluation dataset). If we do it this way, we can indeed compare perplexity from different models, including those with a different tokenization scheme, pretraining corpus, etc. In fact, many datasets like the Hutter Prize does exactly this. Given how correlated perplexity is with downstream performance (not perfectly correlated, of course, but it is quite highly correlated with a lot of downstream performance, even in the case of LLMs), I would like to disagree with this assertion.

5. There are still a number of typos in the paper, so it could benefit from more copy-editing. This is a minor point (primarily because it is a long paper and it is hard to catch all the typos), and I find the paper to be generally clear and well-written. I also have some presentational suggestions:
    - In Section 2.2, is "conditional text generation" basically **sequence-to-sequence generation**? The field used to use this term to refer to any problem that has a sequence-like input and an output, including machine translation, summarization, dialogue systems, etc. If so, might be good to mention the connection.
    - In page 7, "... it is desirable generated dialogue ..." -> "it is desirable **that** ..."
    -  In page 10, "In lack of suitable ..." -> "In **a** lack of ..."
    - In page 16, "while at the same time preserve ..." -> "**preserving**"
    - In page 17, "...allow to rewrite..." -> "allow **us** to rewrite..."
    - In page 18, the reinforcement learning part and why it is difficult might warrant its own subsection, which would improve clarity.
    - In page 24, "... can be mislead ..." -> "**misled**"

---

> ### Author Response · Authors · 2024-12-15
>
> Thank you for your detailed review and acknowledging the strengths of our paper. We would like to address your concerns:
>
> Weaknesses:
> 1. Confusing/Inaccurate Notations:
> Thank you for pointing this out! We notice that different notations and norms have been used in the literature (e.g., some literature summed over sentences rather than tokens to ease the presentation). We will add a footnote to refer readers to these alternative notations.
>
> 2. Belaboured sections:
> Thanks for your constructive suggestion! To your point, we agree that reducing the narratives that the TMLR audience may have been already familiar with could make our paper more clear and concise to this audience. We will update the manuscript and remove the redundant details accordingly.
>
> 3. Points that warrant further discussion:
> What approaches should the field try to get better results in this problem?
>
> Thank you for this pertinent question! We do believe there are several promising approaches in the literature that are worth exploring in more depth for advancing constrained NLG. We discuss them in turn below:
>
> 1) RLHF has become the predominant paradigm for aligning LLMs to human preferences given helpfulness, harmless and safety constraints. Nevertheless, RLHF performance is strongly dependent upon the quality of the reward model, and defining rewards for real-world tasks is non-trivial. Reward functions are fragile and notoriously difficult to specify, particularly for tasks with complex goals [3]. For example, an outstanding challenge is the issue of reward hacking, where LLM policies learn to exploit failures of the reward model and achieve seemingly high rewards without meeting the underlying objectives [4]. Reward hacking is primarily caused by: i) distribution shifts during the RL process, i.e. due to the offline nature of the preference data, generations from the RL policy (as training progresses and the policy moves away from its supervised fine tuned reference) might deviate substantially from the offline examples in the preference dataset the reward model has been trained on, posing an out-of-distribution (OOD) challenge,  and ii) inconsistencies in human preferences leading to label noise as humans often tend to rely on superficial criteria and shallow textual aspects rather than more nuanced indicators when making their labeling decision (for eg., there is low inter-labeler agreement of only 72.6% for InstructGPT). Other limitations of the RLHF framework include training instability, incorrect generalization and the sparsity of feedback [5].
>
> Despite these challenges, we believe that RLHF can play a key role in advancing constrained NLG, conditioned on the design of more robust and reliable reward functions To mitigate reward hacking in particular, it is necessary to have reward models that can reliably score generations despite distribution shifts, are robust to label noise and inconsistencies in human preferences. Preliminary approaches that explore prediction ensembling [7] or weight averaging [4] of multiple reward models are designed to act as regularization preserving only those mechanisms that are invariant across runs, helping reduce reliance on spurious features and memorization of corrupt/noisy training examples. While these approaches may help delay reward hacking to some extent, they do not fully solve the problems of reliability under distribution shifts and robustness to noisy labels. We believe there is a lot of space to explore more efficient solutions for training robust reward models for real-world tasks that accurately reflect human preferences, entirely prevent reward hacking (instead of just delaying it), and generalize to distribution shifts. In addition, enriching preference datasets with diverse generations, possibly via active learning and data augmentation algorithms, can potentially enhance the robustness to preference inconsistencies and improve generalization. Alternatively, filtering noisy examples and suppressing the gradients of samples with high uncertainty
>
> We also think that having RL policies that perform well for a collection of diverse reward functions, as opposed to only one reward function, can help accommodate diverse user preferences and advance NLG with multiple constraints. To this end, multi-objective reinforcement learning (MORL) algorithms can be used to learn Pareto-optimal policies and control over the learnt policies accounting for multiple objectives [9]. For example, instead of relying on a single scalar, incorporating multi-objective reward modeling allows for dynamic control of the tradeoff between diverse user preferences via arithmetic operations in the vector reward space [10].
>
> Finally, we acknowledge that in many real-world settings it may be difficult to accurately specify constraints mathematically. In such situations, it may be possible to learn constraints directly from user provided demonstrations, even when the reward function is unknown [11], [12].

---

> > ### Author Response · Authors · 2024-12-15
> >
> > 2) Mechanistic Interpretability:
> >
> > Recent works aiming to understand the inner workings of LLM models find that these models have internal representations that encode concepts in a disentangled manner; if one can identify which part of the representation subspace corresponds to a given concept, then it its possible manipulate the concepts expressed by the model through algebraic manipulation of the representation [1]. Moreover, the linear representation hypothesis [2] posits that high-level concepts are represented linearly as directions in the representation space. Assuming it is possible to identify these linear representations of concepts, we can then perform linear algebraic operations on the representation space for fine-grained control of the LLM outputs.  Interventions using steering vectors that change the value a concept takes without changing other concepts show that in carefully designed test cases it is possible to change, for eg., the output from English to French by adding a suitable English/French steering vector. Overall, if the linear representation hypothesis is to hold true, this could potentially open up new methods that help advance constrained NLG. Open problems related to this are the identifiability of learned representations, to what extent they capture real-world structure, and what assumptions need to be made about the geometry of the representation space.
> >
> > 3) Causal Interventions / Causal Probing / Inference Time Interventions:
> >
> > Recent work shows that it is possible to perform direct manipulation of computational mechanisms inside LLMs. In particular, causal tracing approaches first identify neuron activations corresponding to a particular concept, then edit the corresponding weights to change model outputs in a desirable way [13], [14]. Nevertheless, the connection between causality-based localization and model editing is still unclear as localization performed by causal tracing is not indicative of which layer to select for model editing [15]. We consider that developing more reliable causal mechanisms for localizing where knowledge is stored inside a neural network, as well as robust editing of model’s internal knowledge to ensure model generated outputs satisfy given constraints can pave the way for advancing constrained NLG.
> >
> > In addition, causal probing can be used to analyze how intervening on latent properties of the model’s representation have an impact on the model outputs [16].  For example, linear probing has been used to predict whether the generated answer will be truthful or not before the answer is generated [17].  While measuring the effectiveness of probing interventions in LLM models is an open research area, we believe that causal probing  is a promising direction to explore whether constraints are present in the generated output before generation even begins. On a related note, inference-time intervention [19] locates directions that correspond to specific concepts (for eg., truthfulness) and shifts model activations along these directions at inference time; this could be useful when used as part of a more comprehensive approach, potentially integrated with causal probing and intervention mechanisms, and a better understanding of the multidimensional geometry of the representation space for complex attributes / constraints.

---

> > > ### Author Response · Authors · 2024-12-15
> > >
> > > 4) Reasoning-based approaches / Constrained Decoding approaches / Rejection Sampling:
> > >
> > > Graph-constrained reasoning [21] aims to connect the unstructured reasoning in LLMs with the structured knowledge found in knowledge graphs (KG), and constrains the LLM decoding process to reasoning paths that encode KG information. Generating KG-grounded paths helps alleviate reasoning issues due to lack of knowledge, mitigates hallucinations and enhances the faithfulness of generated responses. On a related note, constrained chains of reasoning [20] leverage domain knowledge and the causal relations between concepts to construct reasoning chains that improve consistency of the generated responses.
> > >
> > > Constrained decoding approaches enforce adherence to constraints during generation, while ideally minimally intervening on other non-target aspects to avoid misalignment (minimally constrained decoding) [39]. Guided decoding [23] employs an auxiliary evaluation function that captures to what extent partial outputs satisfy given goals, and presents the advantage that it can be combined with search algorithms for obtaining outputs that satisfy specific constraints. For example, inference-time search algorithms such as Monte-Carlo Tree Search (MCTS) applied on top of RLHF fine-tuned models demonstrate that value-guided decoding with MCTS is a crucial component for achieving model steerability and constraint satisfaction (for eg., sentiment steering, toxicity reduction, helpful and harmless chatbots) [22]. While value models trained as byproducts when aligning LLMs to human preferences have only recently been employed as evaluation functions for scoring partial/incomplete sequences and steer LLMs, it would be interesting to use MCTS as a policy optimization operator to search for output sequences with higher rewards that obey given constraints. Historically, MCTS has shown to be an indispensable inference-time component that helped the AlphaGo system reach superhuman performance.
> > >
> > > Overall, we believe that combining reasoning-based approaches with guided decoding and MCTS has a lot of potential to improve the state-of-the-art for constrained NLG, with the added benefit of more faithful and consistent generations with less hallucinations.
> > >
> > > Finally, statistical rejection sampling can be used to discard partial samples that do not meet given constraints; this technique has been widely employed in LLAMA-3 family of models [24], and is found to improve the alignment with human preferences in constraint optimization settings [25]. Best-of-N rejection sampling [26] draws n samples from the LLM, ranks them on the target attribute of interest / desired constraint, and returns the best sample; despite its simplicity, this strategy has been found to be surprisingly effective in practice. Nevertheless, Best-of-N sampling comes at considerable inference cost, and approaches trained to mimic this distribution achieve high win-rates while minimally affecting other off-target aspects of the generation [6]. Such frameworks allow for better control of LLM models and can be used to focus on generating texts that meet desirable on-target constraints, without changing other non-desirable / off-target aspects of the generation.

---

> > > > ### Author Response · Authors · 2024-12-15
> > > >
> > > > 5) Better model architectures:
> > > >
> > > > Long-term Attention Mechanisms / Addressing Instruction Drift and Long Context The mismatch between how LLM models are trained with a next-word prediction objective and how they are used in practice (for eg., for long-term open-ended dialogue with users) leads to inconsistency in their behaviour over long horizons. For example, the Transformer attention mechanism decays over long exchanges, causing chatbots to stray away from prompted behaviour and resulting in instruction drift that degrades the quality of the outputs over lengthy dialogues [27] . Improving instruction stability in LLMs, particularly in long-form conversations, can lead to more stable and robust prompting, improve the performance of current models and their abilities to generate texts accounting for given constraints. Furthemore, while recent LLM models have the ability to take long contexts as input, their performance is often highest when using information that occurs at the beginning or end of the input context, and degrades significantly when models must access relevant information in the middle of long contexts. This indicates that while prompting current models with long input contexts may help in downstream tasks, it also has the potential to decrease performance if important information is included in the middle of the prompt. There is a need for better understanding of how language models use input context and potentially come up with new designs of attention mechanisms that capture information within long input contexts more robustly.
> > > >
> > > >
> > > > Incorporating logical and causal based reasoning in model training Autoregressive LLMs fail to generalize in surprising ways. The reversal curse [30] is one instance where a model trained on a sentence of the form “A is B” will not automatically generalize to the reverse direction “B is A”, therefore failing to deduce the reverse relationship –  this direct impact on the LLM models’ ability to generate constrained texts. Further analysis of LLM generalization using influence functions [31] aiming to understand which training examples contribute to given model behaviour reveals that training examples that match the order (“A precedes B”) are far more influential than examples with reverse order (“B precedes A”). While fine-tuning and data augmentation approaches have been proposed to alleviate this issue, it points to a basic failure of logical deduction in the LLM’s training process. We consider that incorporating logical deduction and causal-based reasoning as part of LLM training could help alleviate such problems and improve the robustness of constrained NLG.
> > > >
> > > > Accounting for constraints / Better guidance during training Classifier guidance [29] introduces an extra trained classifier to guide diffusion model generations in particular desirable directions.  Further improvements of this approach in the form of classifier-free guidance [28] show that it is possible to perform guidance using a pure generative model instead of an additional classifier; the approach jointly trains a conditional and an unconditional diffusion model and combines their score estimates to achieve fine-grained control over the generated outputs. Although these methods have been mainly applied to diffusion models in the image domain, we believe such approaches have a lot of potential to lead to more robust constrained text generation as well.

---

> > > > > ### Author Response · Authors · 2024-12-15
> > > > >
> > > > > 6) Multi-objective / pluralistic benchmarks:
> > > > >
> > > > > Aligning LLM models to pluralistic human values requires the capability to accurately steer models in directions representing a diverse set of human values and perspectives [33]. Steerable pluralism, i.e. faithfully steering LLMs to represent particular attributes or perspectives, plays a key role in personalizing and customizing models to various users or target populations. Nevertheless, current benchmarks for constrained NLG are monistic, focusing on a single objective. Pluralistic benchmarks have more than one objective to maximize (each objective is measured separately) and allow for more explicit trade-offs between constraints at inference time.
> > > > >
> > > > >
> > > > > 7) Hybrid Human-AI Collaborative Approaches: Interactive systems that allow LLM models and their end users to write collaboratively have the potential to enhance constrained text generation by empowering users to choose which (partial) model generations and are in line with their needs and expectations and satisfy given constraints. For example, CoAuthor [32] proposes a collaborative human-AI approach for text generation where a writer and a model takes turns interactively in writing a story and editing it. Such methods empower users with fine-grained control over the model outputs and open up new opportunities in assistive writing in a steerable fashion.
> > > > >
> > > > > Causal inference can play a key role in modeling the human-AI collaboration, and can help answer counterfactual  “what-if” questions on how the outcome of the collaboration would change if humans employed a different text editing/refinement strategy. To this end, causal estimands (Incremental Stylistic Effect) [37] can be used to measure the average impact of infinitesimally shifting a text towards a specific style, for example increasing the degree of politeness or formality.

---

> > > > > > ### Author Response · Authors · 2024-12-15
> > > > > >
> > > > > > Mismatch between constrained NLG benchmarks and real use cases:
> > > > > >
> > > > > > a) Current LLM models are trained on web-scale datasets with minimal curation to ensure data quality. In addition, many pre-training datasets also contain various evaluation benchmarks that are used for assessing and comparing trained models. Due to these factors, interpreting evaluation results must be done with caution accounting for the source pre-training data in determining to what extent current models generalize vs simply memorize training examples. Auditing LLM models for test set contamination via statistical significance tests reveals verbatim contamination, i.e. LLM models are trained directly on the test sets they are evaluated on  [35]. Because of this, LLM performance is likely overestimated on existing benchmarks and their behavior in practice is likely much inferior to reported results. In addition, the inflated evaluation results on benchmarks are not an effective indicator of model generalization abilities and real world performance.  Indeed, recent work shows that emergent abilities of LLMs on many benchmarks are a mirage [36], appearing due the researcher’s choice of evaluation metric rather than due to fundamental changes in model behavior with scale.
> > > > > >
> > > > > > b) Current benchmarks are not representative of the diversity of real-world use cases, and only have limited data coverage they are evaluating models on. Due to lack of holistic evaluations, it may appear that LLMs perform well in controlled environments, however they fail in critical real-world applications, posing safety risks such as perpetuating bias, making unsafe decisions, or being vulnerable to manipulation.
> > > > > >
> > > > > > In addition, recent analysis of state-of-the-art LLM benchmarks finds that they suffer from significant limitations, including biases, difficulties in measuring genuine reasoning, adaptability, implementation inconsistencies, prompt engineering complexity, lack of evaluator diversity, and the overlooking of cultural and ideological norms  [38].
> > > > > >
> > > > > > c) Current benchmarks evaluate model performance across a single dimension and summarize results in the form of a single scalar value which not only offers an incomplete picture of the model performance, but is also misleading when used to compare and rank model submissions. Besides, single-value benchmarks can often lead to “reward-hacking” and exploiting spurious features, such as annotators’ preference for more verbose responses [33].
> > > > > >
> > > > > > d) Existing benchmarks are monistic, focusing on measuring model performance on a single target objective; they are not designed to evaluate model performance on multiple constraints [33]. Pluralistic benchmarks with more than one objective to maximize aim to capture the entire spectrum of model performance across different target objectives, making it feasible to compare and rank different models across a multitude of dimensions. Recent work [33] proposes three types of pluralistic benchmarks:
> > > > > >
> > > > > > - Multi-objective benchmarks: reporting evaluations across all objectives for all solutions;
> > > > > > - Trade-off steerable benchmarks: designed to measure steerability of models and encourage models to trade off between different objectives at inference time;
> > > > > > - Jury-pluralistic benchmarks: model diverse human ratings, allow to explicitly reason over which users or groups models are being aligned to for obtaining fairer outcomes.
> > > > > >
> > > > > > In our view, multi-objective benchmarks and trade-off steerable benchmarks are particularly important for further advancing multi-objective constrained NLG.
> > > > > >
> > > > > > e) Lack of research consensus on how to properly benchmark models and measure scientific progress; continued influx of question-and-answer sets self-claimed as benchmarks; lack of standardized evaluation guidelines [38].

---

> > > > > > > ### Author Response · Authors · 2024-12-15
> > > > > > >
> > > > > > > Areas to revisit that combine nicely with current LLM advances / most promising ways forward:
> > > > > > >
> > > > > > > 1) decoding from RLHF fine tuned policies with constrained decoding and Monte-Carlo Tree Search for ensuring generated texts meet desirable constraints; explore constrained decoding approaches that minimally change non-target aspects of the generation [39].
> > > > > > >
> > > > > > > 2) beyond relying on MCTS during decoding, it would be interesting to explore the use of Monte-Carlo Tree Search as a policy optimization operator during the RLHF training of LLMs; this would ensure the RL trained policy learns to generate texts with desirable constraints during policy training and that its generations are in-distribution for the (constrained) reward model;
> > > > > > >
> > > > > > > 3) better understanding of the multidimensional geometry of the representation space, potentially via unsupervised learning and causal inference methods, aimed to achieve fine-grained model control and model steering in the latent space;
> > > > > > >
> > > > > > > 4) using probing mechanisms to find directions in the latent space that correspond to specific target objectives and ensure texts satisfy desirable constraints before being generated by LLMs;
> > > > > > >
> > > > > > > 5) Diffusion models have been widely used in the field of computer vision for constrained text generation, for example the concept algebra paper [1] shows that concepts are encoded in a disentangled manner as subspaces of some representation space and can be controlled via simple algebraic operations. It would be interesting to adapt diffusion models to constrained text generation for model controllability;
> > > > > > >
> > > > > > > 6) Causal learning and causal inference, causal estimands - incorporating causality and logical deductions for more robust model training and steering, understanding model representations and how changing them in the latent space has a direct impact on the generated output.
> > > > > > >
> > > > > > > 7) Training reward models for real-world tasks and coming up with solid theoretically-grounded methods for overcoming reward hacking (instead of just delaying this phenomenon from occurring by weight-averaging and prediction ensembling).
> > > > > > >
> > > > > > > 8) Revisiting regularization methods in the context of model training / Propose better regularizers that capture human preferences - KL regularization is the standard method in RLHF to ensure the the trained policy does not diverge from its supervised reference, nevertheless the KL can have poor correspondence with aspects of the text that humans consider salient, for example large variations in length are not reflected in the estimated KL [6].
> > > > > > >
> > > > > > > 9) Improve prompting / instruction stability in LLMs, ensure models are following instructions given for generating constrained texts.
> > > > > > >
> > > > > > >
> > > > > > > 4. Perplexity:
> > > > > > >
> > > > > > > We acknowledge that there are ways to compare perplexity across models, but not “directly”. We will change our statement into “how likely a sentence is generated by a given model” is not directly comparable across different models unless properly normalized. Finding the right normalization is a challenge that could potentially improve the evaluation of constrained text generation.
> > > > > > >
> > > > > > > While perplexity evaluations are often correlated with LLM performance, this isn’t aligned with the more conventional models - we believe this can also be framed as an opportunity to revisit the use of perplexity as evaluation metric in the context of LLMs.
> > > > > > >
> > > > > > > 5. In Section 2.2, is "conditional text generation" basically sequence-to-sequence generation?
> > > > > > >
> > > > > > > This is a great point, we will make sure to mention this in the paper!
> > > > > > >
> > > > > > > Typos - thank you for your attention to detail and for catching our typos
> > > > > > >
> > > > > > >
> > > > > > > Thank you very much for your detailed review, we appreciate your suggestions and will incorporate the requested changes in the updated draft of our paper!

---

> > > > > > > > ### Author Response · Authors · 2024-12-15
> > > > > > > >
> > > > > > > > [1] Wang, Zihao, Lin Gui, Jeffrey Negrea, and Victor Veitch. "Concept algebra for (score-based) text-controlled generative models." Advances in Neural Information Processing Systems 36 (2024).
> > > > > > > >
> > > > > > > > [2] Park, Kiho, Yo Joong Choe, and Victor Veitch. "The Linear Representation Hypothesis and the Geometry of Large Language Models." In Forty-first International Conference on Machine Learning (2024).
> > > > > > > >
> > > > > > > > [3] McKinney, Lev, Yawen Duan, David Krueger, and Adam Gleave. "On the fragility of learned reward functions." arXiv preprint arXiv:2301.03652 (2023).
> > > > > > > >
> > > > > > > > [4] Ramé, Alexandre, Nino Vieillard, Léonard Hussenot, Robert Dadashi, Geoffrey Cideron, Olivier Bachem, and Johan Ferret. "Warm: On the benefits of weight averaged reward models." arXiv preprint arXiv:2401.12187 (2024).
> > > > > > > >
> > > > > > > > [5] Chaudhari, Shreyas, Pranjal Aggarwal, Vishvak Murahari, Tanmay Rajpurohit, Ashwin Kalyan, Karthik Narasimhan, Ameet Deshpande, and Bruno Castro da Silva. "RLHF Deciphered: A Critical Analysis of Reinforcement Learning from Human Feedback for LLMs." arXiv preprint arXiv:2404.08555 (2024).
> > > > > > > >
> > > > > > > > [6] Gui, Lin, Cristina Gârbacea, and Victor Veitch. "BoNBoN Alignment for Large Language Models and the Sweetness of Best-of-n Sampling." arXiv preprint arXiv:2406.00832 (2024).
> > > > > > > >
> > > > > > > > [7] Christiano, Paul F., Jan Leike, Tom Brown, Miljan Martic, Shane Legg, and Dario Amodei. "Deep reinforcement learning from human preferences." Advances in neural information processing systems 30 (2017).
> > > > > > > >
> > > > > > > > [8] Qiu, Shuang, Dake Zhang, Rui Yang, Boxiang Lyu, and Tong Zhang. "Traversing pareto optimal policies: Provably efficient multi-objective reinforcement learning." arXiv preprint arXiv:2407.17466 (2024).
> > > > > > > >
> > > > > > > > [9] Liang, Xize, Chao Chen, Jie Wang, Yue Wu, Zhihang Fu, Zhihao Shi, Feng Wu, and Jieping Ye. "Robust preference optimization with provable noise tolerance for llms." arXiv preprint arXiv:2404.04102 (2024).
> > > > > > > >
> > > > > > > > [10] Wang, Haoxiang, Yong Lin, Wei Xiong, Rui Yang, Shizhe Diao, Shuang Qiu, Han Zhao, and Tong Zhang. "Arithmetic control of llms for diverse user preferences: Directional preference alignment with multi-objective rewards." arXiv preprint arXiv:2402.18571 (2024).
> > > > > > > >
> > > > > > > > [11] Lindner, David, Xin Chen, Sebastian Tschiatschek, Katja Hofmann, and Andreas Krause. "Learning safety constraints from demonstrations with unknown rewards." In International Conference on Artificial Intelligence and Statistics, pp. 2386-2394. PMLR, 2024.
> > > > > > > >
> > > > > > > > [12] Malik, Shehryar, Usman Anwar, Alireza Aghasi, and Ali Ahmed. "Inverse constrained reinforcement learning." In International conference on machine learning, pp. 7390-7399. PMLR, 2021.
> > > > > > > >
> > > > > > > > [13] Meng, Kevin, David Bau, Alex Andonian, and Yonatan Belinkov. "Locating and editing factual associations in GPT." Advances in Neural Information Processing Systems 35 (2022): 17359-17372.
> > > > > > > >
> > > > > > > > [14] Meng, Kevin, Arnab Sen Sharma, Alex J. Andonian, Yonatan Belinkov, and David Bau. "Mass-Editing Memory in a Transformer." In The Eleventh International Conference on Learning Representations.
> > > > > > > >
> > > > > > > > [15] Hase, Peter, Mohit Bansal, Been Kim, and Asma Ghandeharioun. "Does localization inform editing? surprising differences in causality-based localization vs. knowledge editing in language models." Advances in Neural Information Processing Systems 36 (2024).
> > > > > > > >
> > > > > > > > [16] Canby, Marc, Adam Davies, Chirag Rastogi, and Julia Hockenmaier. "Measuring the reliability of causal probing methods: Tradeoffs, limitations, and the plight of nullifying interventions." arXiv preprint arXiv:2408.15510 (2024).
> > > > > > > >
> > > > > > > > [17] Joshi, Nitish, Javier Rando, Abulhair Saparov, Najoung Kim, and He He. "Personas as a way to model truthfulness in language models." arXiv preprint arXiv:2310.18168 (2023).
> > > > > > > >
> > > > > > > > [18] Lin, Zizheng, Chunkit Chan, Yangqiu Song, and Xin Liu. "Constrained reasoning chains for enhancing theory-of-mind in large language models." In Pacific Rim International Conference on Artificial Intelligence, pp. 354-360. Singapore: Springer Nature Singapore, 2024.
> > > > > > > >
> > > > > > > > [19] Li, Kenneth, Oam Patel, Fernanda Viégas, Hanspeter Pfister, and Martin Wattenberg. "Inference-time intervention: Eliciting truthful answers from a language model." Advances in Neural Information Processing Systems 36 (2024).
> > > > > > > >
> > > > > > > > [20 Lin, Zizheng, Chunkit Chan, Yangqiu Song, and Xin Liu. "Constrained reasoning chains for enhancing theory-of-mind in large language models." In Pacific Rim International Conference on Artificial Intelligence, pp. 354-360. Singapore: Springer Nature Singapore, 2024.]
> > > > > > > >
> > > > > > > > [21] Luo, Linhao, Zicheng Zhao, Chen Gong, Gholamreza Haffari, and Shirui Pan. "Graph-constrained Reasoning: Faithful Reasoning on Knowledge Graphs with Large Language Models." arXiv preprint arXiv:2410.13080 (2024).
> > > > > > > >
> > > > > > > > [22] Liu, Jiacheng, Andrew Cohen, Ramakanth Pasunuru, Yejin Choi, Hannaneh Hajishirzi, and Asli Celikyilmaz. "Don't throw away your value model! Generating more preferable text with Value-Guided Monte-Carlo Tree Search decoding." In First Conference on Language Modeling. 2024.

---

> > > > > > > > > ### Author Response · Authors · 2024-12-15
> > > > > > > > >
> > > > > > > > > [23] Lu, Ximing, Peter West, Rowan Zellers, Ronan Le Bras, Chandra Bhagavatula, and Yejin Choi. "NeuroLogic Decoding:(Un) supervised Neural Text Generation with Predicate Logic Constraints." In Proceedings of the 2021 Conference of the North American Chapter of the Association for Computational Linguistics: Human Language Technologies, pp. 4288-4299. 2021.
> > > > > > > > >
> > > > > > > > > [24] Dubey, Abhimanyu, Abhinav Jauhri, Abhinav Pandey, Abhishek Kadian, Ahmad Al-Dahle, Aiesha Letman, Akhil Mathur et al. "The llama 3 herd of models." arXiv preprint arXiv:2407.21783 (2024).
> > > > > > > > >
> > > > > > > > > [25] Liu, Tianqi, Yao Zhao, Rishabh Joshi, Misha Khalman, Mohammad Saleh, Peter J. Liu, and Jialu Liu. "Statistical Rejection Sampling Improves Preference Optimization." In The Twelfth International Conference on Learning Representations, 2024.
> > > > > > > > >
> > > > > > > > > [26] Stiennon, Nisan, Long Ouyang, Jeffrey Wu, Daniel Ziegler, Ryan Lowe, Chelsea Voss, Alec Radford, Dario Amodei, and Paul F. Christiano. "Learning to summarize with human feedback." Advances in Neural Information Processing Systems 33 (2020): 3008-3021.
> > > > > > > > >
> > > > > > > > > [27] Li, Kenneth, Tianle Liu, Naomi Bashkansky, David Bau, Fernanda Viégas, Hanspeter Pfister, and Martin Wattenberg. "Measuring and controlling instruction (in) stability in language model dialogs." In First Conference on Language Modeling. 2024.
> > > > > > > > >
> > > > > > > > > [28] Ho, Jonathan, and Tim Salimans. "Classifier-free diffusion guidance." arXiv preprint arXiv:2207.12598 (2022).
> > > > > > > > >
> > > > > > > > > [29] Dhariwal, Prafulla, and Alexander Nichol. "Diffusion models beat gans on image synthesis." Advances in neural information processing systems 34 (2021): 8780-8794.
> > > > > > > > >
> > > > > > > > > [30] Berglund, Lukas, Meg Tong, Max Kaufmann, Mikita Balesni, Asa Cooper Stickland, Tomasz Korbak, and Owain Evans. "The reversal curse: Llms trained on" a is b" fail to learn" b is a"." arXiv preprint arXiv:2309.12288 (2023).
> > > > > > > > >
> > > > > > > > > [31] Grosse, Roger, Juhan Bae, Cem Anil, Nelson Elhage, Alex Tamkin, Amirhossein Tajdini, Benoit Steiner et al. "Studying large language model generalization with influence functions." arXiv preprint arXiv:2308.03296 (2023).
> > > > > > > > >
> > > > > > > > > [32] Lee, Mina, Percy Liang, and Qian Yang. "Coauthor: Designing a human-ai collaborative writing dataset for exploring language model capabilities." In Proceedings of the 2022 CHI conference on human factors in computing systems, pp. 1-19. 2022.
> > > > > > > > >
> > > > > > > > > [33] Sorensen, Taylor, Jared Moore, Jillian Fisher, Mitchell L. Gordon, Niloofar Mireshghallah, Christopher Michael Rytting, Andre Ye et al. "Position: A Roadmap to Pluralistic Alignment." In Forty-first International Conference on Machine Learning.
> > > > > > > > >
> > > > > > > > > [34] Liu, Nelson F., Kevin Lin, John Hewitt, Ashwin Paranjape, Michele Bevilacqua, Fabio Petroni, and Percy Liang. "Lost in the middle: How language models use long contexts." Transactions of the Association for Computational Linguistics 12 (2024): 157-173.
> > > > > > > > >
> > > > > > > > > [35] Oren, Yonatan, Nicole Meister, Niladri S. Chatterji, Faisal Ladhak, and Tatsunori Hashimoto. "Proving Test Set Contamination in Black-Box Language Models." In The Twelfth International Conference on Learning Representations.
> > > > > > > > >
> > > > > > > > > [36] Schaeffer, Rylan, Brando Miranda, and Sanmi Koyejo. "Are emergent abilities of large language models a mirage?." Advances in Neural Information Processing Systems 36 (2024).
> > > > > > > > >
> > > > > > > > > [37] Zhang, Bohan, Yixin Wang, and Paramveer S. Dhillon. "Causal Inference for Human-Language Model Collaboration." arXiv preprint arXiv:2404.00207 (2024).
> > > > > > > > >
> > > > > > > > > [38] ​​McIntosh, Timothy R., Teo Susnjak, Nalin Arachchilage, Tong Liu, Paul Watters, and Malka N. Halgamuge. "Inadequacies of large language model benchmarks in the era of generative artificial intelligence." arXiv preprint arXiv:2402.09880 (2024).
> > > > > > > > >
> > > > > > > > > [39] Beurer-Kellner, Luca, Marc Fischer, and Martin Vechev. "Guiding LLMs The Right Way: Fast, Non-Invasive Constrained Generation." arXiv preprint arXiv:2403.06988 (2024).

---

> ### Comment · Reviewer_ndnQ · 2025-01-08
> **Thank You for the Response**
>
> Thank you for the comprehensive authors' response. The future directions to explore more as a field sound good to me, and would increase the value of the paper by outlining some promising areas to focus on. I particularly like the mechanistic interpretability & multi-objective RLHF directions. Based on the authors' response, my concerns regarding (i) what the field should focus on in this research direction; (ii) mismatch between constrained NLG benchmarks & real-world applications; (iii) promising approaches to try from the modelling perspective; and (iv) perplexity comparison across different tokenization schemes have been addressed. I look forward to seeing these in the final version of the paper.
>
> My remaining concern is regarding the notation. The authors' response states that "We notice that different notations and norms have been used in the literature (e.g., some literature summed over sentences rather than tokens to ease the presentation). We will add a footnote to refer readers to these alternative notations.". The objective function as it is written on the initial manuscript is still not correct. Without a summation over the different tokens $i$, for each sentence $\mathbf{x}^{k}$, we are basically only training the model to predict a single token $x_i$ based on the prefix. The correct objective function is what I mentioned earlier, where we properly sum over the different token positions, $\ell(D) = - \sum_{k=1}^{|D|} \sum_{i=1}^{N_k} \log p_{\theta}(x_{i}^{k} \mid \mathbf{x}_{<i}^{k})$ .
>
> If the authors would like to use an objective function that simply sums over the sentences, then alternatively the objective function can be rewritten as $\ell(D) = - \sum_{k=1}^{|D|}  \log p_{\theta}(\mathbf{x}^{k} )$ (i.e. we sum the probability of generating **each sentence** $\mathbf{x}^{k}$ in the dataset $D$, which can then be decomposed into the probability of generating each token based on its prefix / left context.
>
> Either of these 2 objective functions (summing over each sentence, and then summing over the probability of generating each token in that sentence based on the prefix, **or** summing over the probability of generating each sentence) would be correct, but the current objective function as it is written implies that, for each sentence $\mathbf{x}^{k}$, we are only training the model to predict a single token $x_i$ (so we'd need the summation over all token positions, if it is written in this way).

---

> > ### Author Response · Authors · 2025-01-09
> >
> > Thank you for your response and further details. We will update the notation for the objective function as per your suggestion in the updated manuscript!

---

### Decision · Action_Editor_JM1D · 2025-01-15

**Recommendation:** Accept with minor revision

**Comment:**

As with many survey papers, it is difficult to define exactly whether the "claim" of providing a survey is supported. In general, I think it is in this case, although there are several revisions that would make the paper stronger.

Reviewer ndnQ gives an extremely detailed review with some very concrete recommendations. Please include the discussion in the paper and make the required notation fixes. I would appreciate it if you can highlight the changes for ease of reviewing.

Reviewer Umg2 complains about the datedness of cites. The citations to older work are, in my view, commendable, and not a problem at all.  However, I think the survey feels a bit dated in its orientation at times (that is, preferring older work and older topics to the exclusion of newer ones). For one thing, there is relatively little treatment of the most canonical approach for constrained NLG these days, which is prompting.

Section 5.2 discusses fine-tuning approaches including modern LLMs, and section 7 discusses some relevant datasets like IFEval (which is a modern take on most of the constrained generation literature like CommonGen).  However, the future work directions do not make enough mention of this in my view:
* Multiple constraint satisfaction doesn't mention prompting, yet this is the go-to solution these days. Can prompted models satisfy multiple constraints? At what LLM scales? Benchmarks like IFEval, FollowBench, etc. have sought to answer this question and contain some evaluation. This should be mentioned.
* The future direction on "rule constraints" would do well to mention this as well: does prompting models for lexical constraints work? If so, do we need methods that enforce these constraints in a hard way? If not, where would c
onstrained NLG methods be most useful?
* Finally, dynamically-defined constraints and generative reasoning both tie in with modern LLM methodology (and the latter is one of the biggest topics in the field, with whole surveys dedicated to it like https://arxiv.org/html/2407.11511v1 ). I think these sections should be a bit better contextualized: this survey is neither introducing these nor really calling special attention to them, as they are already well-studied problems.

Nevertheless, the survey provides a substantial treatment of the subject and will be valuable to readers who aren't familiar with this area.

Requested changes:

1. ndnQ's changes
2. Provide better contextualization of the future work as requested above
3. Fix the references style; citep should be used almost everywhere, so the cites look like (Krause et al., 2020).

**Audience:**

Yes, this is a topic within scope for TMLR and of interest to its audience.

**Claims And Evidence:**

Yes. This paper offers a survey of constrained NLG, and despite some flaws, it generally delivers on providing this.